# TIP30 counteracts cardiac hypertrophy and failure by inhibiting translational elongation

Andrea Grund[1,2], Malgorzata Szaroszyk[1], Mortimer Korf-Klingebiel[1], Mona Malek Mohammadi[1,2], Felix A Trogisch[2], Ulrike Schrameck[1], Anna Gigina[1], Christopher Tiedje[3], Matthias Gaestel[3], Theresia Kraft[4], Jan Hegermann[5], Sandor Batkai[6], Thomas Thum[6,7], Andreas Perrot[8], Cris dos Remedios[9], Eva Riechert[10], Mirko Völkers[10,11], Shirin Doroudgar[10,11], Andreas Jungmann[10], Ralf Bauer[10], Xiaoke Yin[12], Manuel Mayr[12], Kai C Wollert[1,7], Andreas Pich[13], Hua Xiao[14], Hugo A Katus[10,11], Johann Bauersachs[1,7], Oliver J Müller[10,11,15,†] & Joerg Heineke[1,2,7,11,†,*]

## Abstract

Pathological cardiac overload induces myocardial protein synthesis and hypertrophy, which predisposes to heart failure. To inhibit hypertrophy therapeutically, the identification of negative regulators of cardiomyocyte protein synthesis is needed. Here, we identified the tumor suppressor protein TIP30 as novel inhibitor of cardiac hypertrophy and dysfunction. Reduced TIP30 levels in mice entailed exaggerated cardiac growth during experimental pressure overload, which was associated with cardiomyocyte cellular hypertrophy, increased myocardial protein synthesis, reduced capillary density, and left ventricular dysfunction. Pharmacological inhibition of protein synthesis improved these defects. Our results are relevant for human disease, since we found diminished cardiac TIP30 levels in samples from patients suffering from end-stage heart failure or hypertrophic cardiomyopathy. Importantly, therapeutic overexpression of TIP30 in mouse hearts inhibited cardiac hypertrophy and improved left ventricular function during pressure overload and in cardiomyopathic mdx mice. Mechanistically, we identified a previously unknown anti-hypertrophic mechanism, whereby TIP30 binds the eukaryotic elongation factor 1A (eEF1A) to prevent the interaction with its essential co-factor eEF1B2 and translational elongation. Therefore, TIP30 could be a therapeutic target to counteract cardiac hypertrophy.

**Keywords** cardiac hypertrophy; cardiomyopathy; heart failure; protein synthesis; translational elongation

**Subject Category** Cardiovascular System

## Introduction

Pathological cardiac hypertrophy, which predisposes to the development of heart failure, frequently develops as consequence of ventricular pressure overload, myocardial infarction or due to inherited cardiomyopathy (Heineke & Molkentin, 2006; Hill & Olson, 2008). It is associated with decreased cardiac function, increased cardiomyocyte size, interstitial fibrosis, and capillary rarefaction (Hein et al, 2003). Many signaling proteins were identified that act in concert to trigger transcription of a pro-hypertrophic gene program (Heineke & Molkentin, 2006; Hill & Olson, 2008). This gene program entails mainly qualitative changes in gene expression, but does not account for the quantitative changes during cardiac growth, which are characterized by strong accumulation of newly synthesized proteins that can lead to enlargement of the heart by more than 50% (Nagatomo et al, 1999; McDermott et al, 2012). The strong increase in cardiac protein content

1 Department for Cardiology and Angiology, Hannover Medical School, Hannover, Germany
2 Department of Cardiovascular Research, European Center for Angioscience (ECAS), Medical Faculty Mannheim, University of Heidelberg, Mannheim, Germany
3 Institute of Cell Biochemistry, Hannover Medical School, Hannover, Germany
4 Institute for Molecular and Cellphysiology, Hannover Medical School, Hannover, Germany
5 Research Core Unit Electron Microscopy, Hannover Medical School, Hannover, Germany
6 Institute of Molecular and Translational Therapeutic Strategies (IMTTS), Hannover Medical School, Hannover, Germany
7 Cluster of Excellence Rebirth, Hannover Medical School, Hannover, Germany
8 Experimental and Clinical Research Center, A Joint Cooperation of Max-Delbrück Center for Molecular Medicine and Charité-Universitätsmedizin Berlin, Berlin, Germany
9 Sydney Heart Bank, University of Sydney, Sydney, NSW, Australia
10 Department of Cardiology, Angiology and Pneumology, Medical Faculty of Heidelberg, University of Heidelberg, Heidelberg, Germany
11 DZHK (German Centre for Cardiovascular Research), Partner Site Heidelberg/Mannheim, Heidelberg, Germany
12 King's British Heart Foundation Centre, King's College London, London, UK
13 Core Unit Proteomics, Hannover Medical School, Hannover, Germany
14 Department of Physiology, Michigan State University, East Lansing, MI, USA
15 Department of Internal Medicine III, Cardiology, Angiology and Intensive Care Medicine, Universitätsklinikum Schleswig-Holstein, Kiel, Germany
*Corresponding author. Tel: +49-621-383-71855; Fax: +49-621-383-71851; E-mail: Joerg.Heineke@medma.uni-heidelberg.de
†These authors contributed equally to this work

mainly results from enhanced protein synthesis within the first 1–5 days of pressure overload with or without a significant decrease in protein degradation (Nagatomo *et al*, 1999; McDermott *et al*, 2012). After 10–14 days, hypertrophy reaches its maximum and a new steady state is attained, in which protein synthesis equals protein degradation and cardiac mass remains stable (Nagatomo *et al*, 1999). mTOR as catalytic subunit of the mTOR containing multiprotein complex 1 (mTORC1) promotes cardiac protein synthesis and hypertrophy mainly by fostering translational initiation (Laplante & Sabatini, 2012). mTORC1 inhibition by rapamycin or its partial deletion in zebrafish improves cardiac function, although its complete genetic abrogation in cardiomyocytes induces cardiomyopathy (Shioi *et al*, 2003; McMullen *et al*, 2004; Ma & Blenis, 2009; Ding *et al*, 2011; Zhang *et al*, 2011b). As overgrowth of the myocardium is associated with poor prognosis during disease (Levy *et al*, 1990), the identification of currently largely undefined endogenous negative regulators of hypertrophy at the level of cardiomyocyte protein synthesis might reveal interesting future therapeutic targets, especially when their abundance is dysregulated in failing hearts.

Protein synthesis is a tightly regulated process that is initiated at the start codon by the 80S ribosome and continues into elongation wherein the peptide chain increases its length cyclically one amino acid at a time (Sasikumar *et al*, 2012). Translational elongation is catalyzed by the eukaryotic translation elongation factor 1A (eEF1A), which in its active GTP-bound form binds and delivers amino acid loaded tRNAs to the A-site of the ribosome. By formation of the correct codon–anticodon pair between tRNA and mRNA, a conformational change in the ribosome leads to GTP hydrolysis and release of then inactive, GDP-bound eEF1A. GDP needs to be actively exchanged for GTP by the guanine nucleotide exchange factor (GEF) eEF1B2, in order to enable eEF1A to participate in another round of elongation.

Here, we characterized the 30 kDa protein TIP30 (also termed Htatip2) as inhibitor of mRNA translation and cardiac hypertrophy and revealed that it protects against heart failure during pathological stimulation. TIP30 is ubiquitously expressed and is acting as tumor suppressor, since reduced TIP30 levels were found in human cancers and were related to enhanced tumor growth and metastasis formation (Shtivelman, 1997; Ito *et al*, 2003; Zhao *et al*, 2007; Li *et al*, 2009). Moreover, homozygous ($Tip30^{-/-}$, KO) and heterozygous ($Tip30^{+/-}$, Het) $Tip30$ knock-out mice develop malignant tumors starting at 18–20 months of age (Ito *et al*, 2003; Li *et al*, 2013; Chen *et al*, 2014). The role of TIP30 in the heart, however, had so far not been analyzed. TIP30 is well conserved across species, and crystallographic analyses suggest binding of NADPH, but found enzymatic activity of TIP30 to be very unlikely (El Omari *et al*, 2005). Instead, it was suggested that TIP30 might play a regulatory role by mediating protein interactions (El Omari *et al*, 2005; Nakahara *et al*, 2009). Accordingly, we demonstrate here that TIP30 interacts with eEF1A to prevent association with its co-factor eEF1B2, thereby blocking translational elongation and cardiomyocyte hypertrophy.

## Results

### TIP30 deficiency facilitates cardiac hypertrophy and failure

To assess the functional role of TIP30 during cardiac overload, we subjected heterozygous (Het, with a 50–60% reduction of cardiac TIP30) and homozygous $Tip30$ knock-out (KO, completely deficient of TIP30) as well as wild-type (WT) mice to sham or transverse aortic constriction (TAC) surgery (Fig 1A). While no phenotypic differences were noted after sham operation, Het and KO mice developed more cardiac hypertrophy (i.e., increased heart weight/tibia length ratio, HW/TL; Fig 1B) 6 weeks after TAC surgery. Het, but not KO or WT mice exerted enhanced pulmonary congestion (increased lung weight/TL; Fig 1C) as sign of cardiac dysfunction after TAC. Accordingly, echocardiography revealed decreased cardiac systolic function (fractional area change) in Het mice and increased cardiac dilation (LVEDA) in Het and KO mice versus WT mice 6 weeks after TAC (Fig 1D and E). Increased dilation and wall thickness of the left ventricle (indicative of enhanced hypertrophy), as well as cardiac dysfunction, were already observed in Het (but not KO) versus WT mice 2 weeks after TAC in echocardiography (Fig EV1A–D). Because Het mice therefore showed a more prominent phenotype than KO mice, we carried out most of the following experiments in Het in comparison with WT mice. To rule out principal differences in the degree of pressure overload after TAC between both genotypes, we conducted Doppler measurements of right versus left carotid artery blood flow. The results indicated that a similar degree of left ventricular pressure overload was reached in Het and WT mice 2 days after TAC versus sham surgery (Fig EV1E).

Direct analysis of left ventricular pressure development by catheterization revealed decreased left ventricular contractility (d$P$/d$t$ max), relaxation (d$P$/d$t$ min), and systolic pressure in Het versus WT mice during pressure overload (Figs 1F and G, and EV1F). Single cardiomyocyte contractility, however, was not different between WT and Het cardiomyocytes after TAC at three different pacing rates (Fig EV1G). Six weeks after TAC surgery, we found a similarly reduced expression of α-myosin heavy chain (α-MHC), but significantly more increased β-MHC expression in the myocardium of Het mice (Fig 1H and I). Cardiac fibrosis was not different between the experimental groups (Fig 1J and K), and accordingly, the number of PDGFRα-positive cardiac fibroblasts was also not changed between them (Fig EV1H and I). In line with the increased HW/TL ratio, enlarged cardiomyocytes were found in Het versus WT mice after TAC (Fig 1L). This augmented cardiomyocyte growth was not accompanied by growth of the cardiac micro-vasculature, since we detected a prominent reduction of the capillary/cardiomyocyte ratio selectively in Het mice after TAC (Fig 1M and N). As capillary rarefaction during pressure overload is known to be maladaptive, it might at least partially contribute to cardiac dysfunction in Het mice during TAC (Heineke *et al*, 2007; Sano *et al*, 2007; Heineke, 2012). In contrast to capillary density, the rate of apoptotic (i.e., cleaved caspase 3 positive) cardiomyocytes was not different between WT and Het mice (Fig EV1J). Profiling of the myocardium of WT and Het mice after TAC by electron microscopy excluded gross ultrastructural defects in these mice (Fig EV1K).

We next analyzed whether increased hypertrophy in $Tip30$ Het mice was the result of TIP30 deficiency primarily in cardiomyocytes, as these mice have systemically reduced TIP30 levels. Aggravated cardiac hypertrophy and pulmonary congestion in Het mice after TAC were reversed upon mild cardiomyocyte specific overexpression of TIP30 by a highly cardiomyocyte selective troponin T promoter-dependent AAV9 vector (AAV9-TropT-TIP30; Figs 1O–R, and EV1L and M; Werfel *et al*, 2014), indicating that lack of TIP30

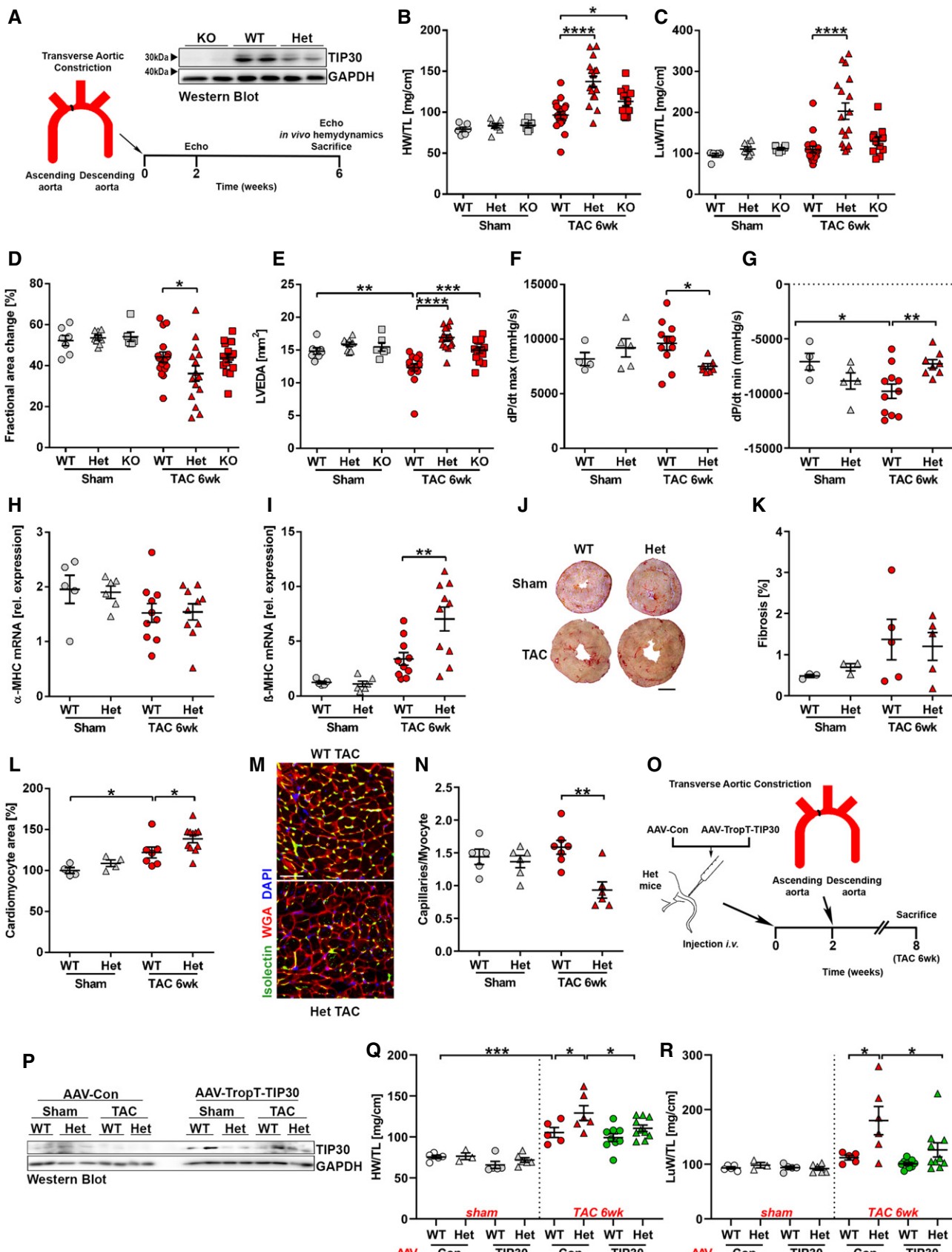

Figure 1.

**Figure 1. TIP30 deficiency results in enhanced cardiac hypertrophy during pathological overload.**

A    Schematic representation of the study design and Western blot analysis for TIP30 and GAPDH in hearts from TIP30 wild-type (WT), heterozygous (Het), and homozygous knock-out (KO) mice under basal conditions.

B–I   Quantification of heart weight (HW)/tibia length (TL) ratio (B), lung weight (LuW/TL) ratio (C), echocardiographic fractional area change (D) and left ventricular end-diastolic area (LVEDA; E), d$P$/d$t$ max and d$P$/d$t$ min (Millar catheter; F, G), and α-MHC and β-MHC transcript abundance (H, I). $N$ = 4–18 mice/group, all 6 weeks after TAC or sham surgery. *$P$ < 0.05, **$P$ < 0.01, ***$P$ < 0.001 and ****$P$ < 0.0001. One-way ANOVA with Sidak's multiple comparisons test.

J, K  Representative images of Sirius red-stained heart sections (scale bar: 1 mm) (J) and fibrosis quantification (K) of indicated mice 6 weeks after TAC or sham surgery. $N$ = 3-5 mice/group.

L     Quantification of cardiomyocyte area of isolated adult cardiac myocytes ($N$ = 4–11 mice/group) of indicated mice 6 weeks after TAC or sham surgery. *$P$ < 0.05. One-way ANOVA with Sidak's multiple comparisons test.

M, N  Microscopy images of heart sections of indicated mice 6 weeks after TAC surgery stained for isolectin B4 (green) and WGA (red, M) and quantification of capillaries per myocyte (N). ($N$ = 5–7 mice/group, scale bar: 50 μm). **$P$ < 0.01. One-way ANOVA with Sidak's multiple comparisons test.

O     Schematic representation of AAV-TopT-TIP30 study design.

P     Western blot analysis for TIP30 and GAPDH in hearts from TIP30 wild-type (WT) and heterozygous (Het) after AAV-TropT-TIP30 or AAV-control (AAV-Con) injection followed by 6 weeks of TAC surgery.

Q, R  Quantification of HW/TL ratio (Q) and LuW/TL (R) ratio in AAV-Con or AAV-TropT-TIP30 treated *Tip30* heterozygous (Het) or WT mice 6 weeks after TAC or sham surgery ($N$ = 5–11 mice/group). *$P$ < 0.05, ***$P$ < 0.001. One-way ANOVA with Sidak's multiple comparisons test.

Data information: Data are shown as mean ± SEM.
Source data are available online for this figure.

in cardiomyocytes of Het mice is predominantly contributing to the observed phenotype in these mice.

Next, we assessed the impact of TIP30 on heart growth during homeostatic conditions: With increasing age, 7-month-old *Tip30* Het mice developed enhanced hypertrophy (indicated by an increased HW/TL ratio, an increased wall thickness in echocardiography, and an increased cardiomyocyte area in histological sections) versus WT mice without any additional stress stimulation, but this was not associated with cardiac dysfunction or dilatation (Fig EV2A–G).

### Reduced myocardial mTORC1 activation in *Tip30* Het mice

Because we observed enhanced cardiac growth in *Tip30* Het mice during pathological stimulation and with increasing age, we analyzed different cell growth-related signaling pathways in the myocardium of WT and Het mice 6 weeks after sham or TAC surgery. We detected a decreased activation of pro-hypertrophic mTORC1 in Het mice after TAC, which was reflected by markedly reduced levels of p70S6Kinase phosphorylation, reduced mTOR phosphorylation and to a lesser extent 4E-BP1 phosphorylation (Fig EV3A and B). A reduction in p70S6K phosphorylation was even visible in Het versus WT mice after sham surgery. The decrease in mTORC1 activation in the myocardium of Het mice was counterintuitive, because these mice showed more heart growth, and we

therefore propose that this was a secondary phenomenon to limit the increased protein synthesis that we detected in the myocardium of Het mice (see below). In addition, the decreased activation of ERK1/2 as well as the increased AMPK activation in the myocardium of Het mice after TAC might directly contribute to reduced mTOR activity in Het mice (Wullschleger *et al*, 2006). The activation and/or abundance of other growth-signaling pathways (p38/JNK-MAPK, eEF2, Akt) was not significantly changed between WT and Het mice.

### TIP30 overexpression restricts cardiac hypertrophy and improves heart function

Neonatal rat cardiomyocytes (NRCM) are widely used as model system to study cardiac hypertrophy. Stimulation of NRCM with the pro-hypertrophic growth factors phenylephrine (PE), fetal bovine serum (FBS), or endothelin-1 (ET-1) led to a mild (about twofold) induction of TIP30 protein levels (Appendix Fig S1A). Since reduced *Tip30* expression in Het mice led to increased cardiac hypertrophy, we wanted to assess whether TIP30 overexpression could inhibit this response. We used a recombinant adenovirus to overexpress TIP30 (Ad.TIP30) in NRCM (Fig 2A). When stimulated with ET-1 or PE, Ad.TIP30 markedly reduced the increase in cell size compared to Ad.Control-treated NRCM and the same trend was observed during FBS stimulation (Fig 2B). Overexpression of TIP30 also

**Figure 2. TIP30 overexpression inhibits cardiac hypertrophy.**

A    Western blot for TIP30 and GAPDH in neonatal rat cardiomyocytes (NRCM) after adenoviral transduction with Ad.Control (Ad.Con) or Ad.TIP30.

B–E   Quantification of cardiomyocyte area, $N$ = 6–8 samples/group (B), protein/DNA ratio, $N$ = 9 samples/group (C), Acta1 mRNA transcript abundance, $N$ = 3 samples/group (D), and cell death with a 7-AAD assay, $N$ = 7 samples/group (E) in NRCM transduced with Ad.Con or Ad.TIP30 and stimulated as indicated. ET-1: endothelin-1, FBS: fetal bovine serum, PE: phenylephrine. *$P$ < 0.05, **$P$ < 0.01, ***$P$ < 0.001 and ****$P$ < 0.0001. One-way ANOVA with Sidak's multiple comparisons test.

F     Schematic representation of AAV-MLC-TIP30 study design.

G     Western blot for TIP30 and Actin in mouse hearts with AAV9 mediated overexpression of TIP30 (AAV-TIP30) or from mice treated with a control AAV9 construct (AAV-Con) followed by 2 weeks of TAC surgery.

H, I  Quantification of HW/TL ratio, $N$ = 8–13 mice/group (H) and cardiomyocyte area, $N$ = 4–5 mice/group (I) 2 weeks after sham or TAC surgery in AAV-Con or AAV-TIP30-treated C57BL/6 WT mice. *$P$ < 0.05. One-way ANOVA with Sidak's multiple comparisons test.

J     Serial echocardiography with quantification of echocardiographic fractional area change 2, 4, and 6 weeks after sham or TAC surgery in AAV-Con or AAV-TIP30-treated C57BL/6 WT mice ($N$ = 10–14 mice/group and time point). **$P$ < 0.01. One-way ANOVA with Sidak's multiple comparisons test.

Data information: Data are shown as mean ± SEM.
Source data are available online for this figure.

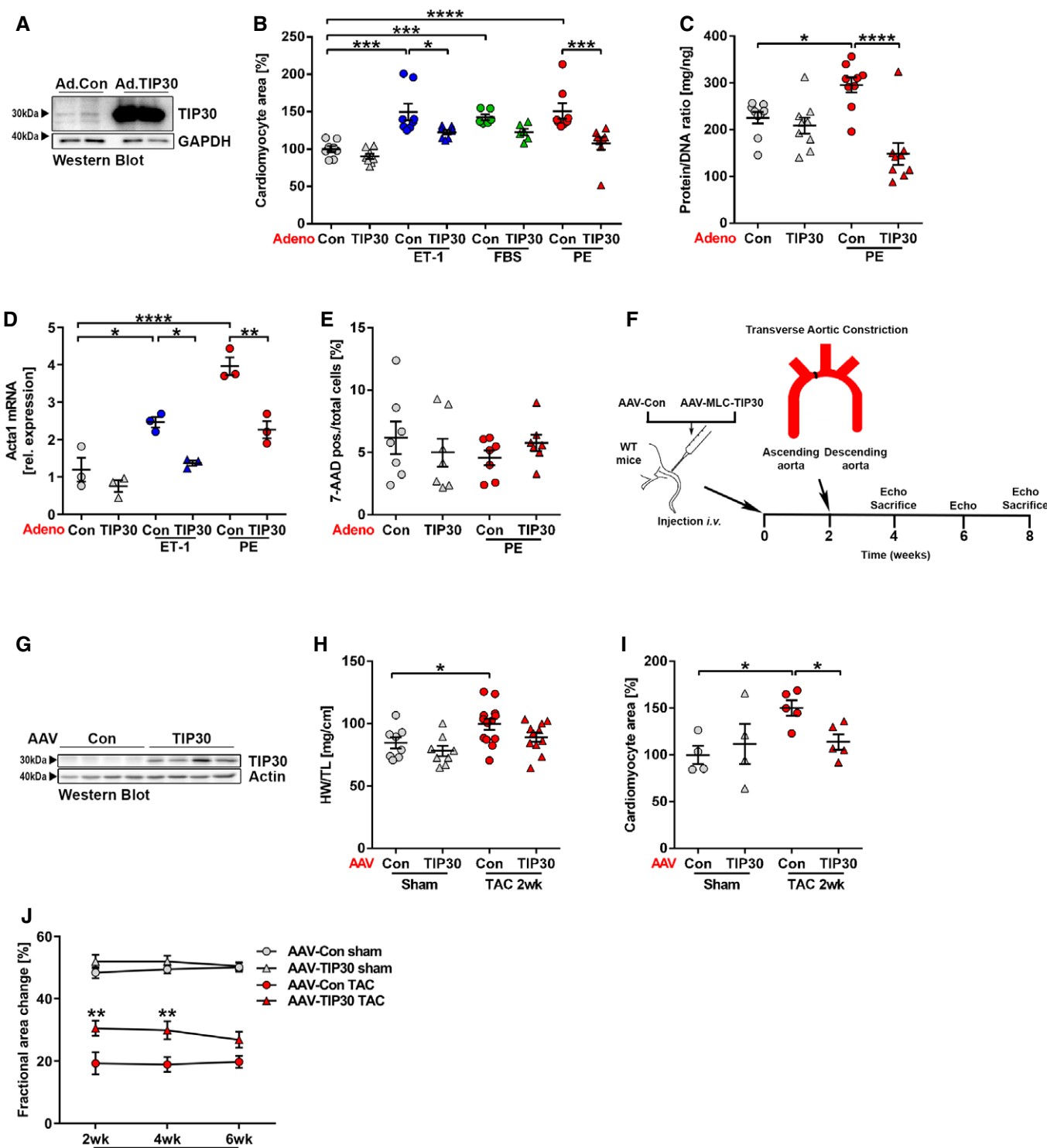

Figure 2.

blunted the increase in cellular protein content (measured as protein/DNA ratio) in response to PE and the expression of the hypertrophic marker gene *Acta1* during ET-1 or PE stimulation (Fig 2C and D). Because TIP30 overexpression was reported to induce cell death in cancer models (Xiao *et al*, 2000), we analyzed

whether TIP30 acts similarly in cardiomyocytes. However, cardiomyocyte death (assessed by 7-AAD staining) was not enhanced by TIP30 overexpression with or without PE stimulation (Fig 2E). Next, we analyzed the effects of cardiac TIP30 overexpression (via AAV9 vector, containing a modified myosin-light chain

promoter) in WT mouse hearts *in vivo* (Fig 2F and G). In line with the results from NRCM, transduction with AAV-TIP30 blunted the increase in heart and cardiomyocyte hypertrophy visible in AAV-control-treated mice in response to 2 weeks of TAC (Fig 2H and I). Serial echocardiography revealed a sustained improvement of systolic cardiac function in AAV-TIP30-treated mice 2, 4, and 6 weeks after TAC surgery (Fig 2J). Interrogation of growth signaling in hearts or isolated cardiomyocytes with TIP30 overexpression did not reveal any significant effects during hypertrophic stimulation (Appendix Fig S1B and C).

### TIP30 binds eEF1A

In order to elucidate the molecular mechanisms that underlie the anti-hypertrophic effect of TIP30, we screened for TIP30 interacting proteins by GST-pulldown assay from NRCM. A number of proteins enriched in the GST-TIP30 versus the GST-pulldown (identified by mass spectrometry) were at least partially associated with the translational apparatus: nucleolin (Ncl), eEF1A1, heterogeneous nuclear ribonucleoprotein (hnRNP)A2/B1, and ribosomal protein (Rp)S3a (Appendix Table S1). We verified the interaction of these proteins with TIP30 in a GST-pulldown assay with GST-TIP30 constructs of different length (Fig 3A). All identified proteins interacted with full-length TIP30. While the first N-terminal 230 amino acids of TIP30 (total length: 242 amino acids) were necessary for interaction with Ncl and Rps3a, only the first 50 amino acids were needed to interact with eEF1A1, and full-length TIP30 was required for binding to hnRNPA2/B1.

We subsequently focused on the interaction between eEF1A1 and TIP30. We hypothesized that TIP30 exerts its anti-hypertrophic effects by interfering with eEF1A1 and by inhibiting protein synthesis during translational elongation. The interaction of both proteins was verified by co-immunoprecipitation of GST-tagged TIP30 full-length protein with Myc-tagged eEF1A1 (Fig 3B). Endogenous TIP30 and eEF1A1 partially co-localized in cardiomyocytes as shown by immunofluorescence staining (Fig 3C). We employed a proximity ligation assay to more directly assess the interaction of both proteins *in situ*. As demonstrated in Fig 3D, endogenous TIP30 interacted with eEF1A1 (each interaction is indicated by a red dot). Endogenous TIP30 and eEF1A1 were also co-immunoprecipitated from NRCM (Fig 3E). A pulldown assay with GST-eEF1A1 showed that the N-terminal 25 amino acids of TIP30 (as part of its NADPH binding domain) are necessary for eEF1A1 binding (Fig 3F–H). In turn, a pulldown assay with GST-TIP30 revealed that recombinant eEF1A1 lacking the middle domain, which is important for eEF1B binding, could not bind TIP30, while domains one or three were not essential for binding (Fig 3I).

Beside the ubiquitous eEF1A1, the heart also expresses its isoform eEF1A2 (Chambers *et al*, 1998). While eEF1A1 mRNA and protein were reduced after birth in the heart as previously reported, we found that it is strongly re-induced in the adult myocardium in response to TAC (Fig 3J and K). In contrast, eEF1A2 mRNA and protein were markedly induced in the adult compared to neonatal hearts, but remained unchanged after TAC. A GST-TIP30 pulldown revealed that TIP30 also interacts with eEF1A2, the isoform of eEF1A mainly expressed in adult myocardium (Fig 3L).

### TIP30 inhibits the interaction of eEF1A1 with its co-factor eEF1B2

Next, we elucidated the consequences of the interaction between TIP30 and eEF1A. eEF1A binds tRNAs and delivers amino acids to the A-site of the ribosome during protein synthesis. To fulfill this function, eEF1A binds eEF1B2 and exists in its GTP-bound form (Pittman *et al*, 2009). Inactive, GDP-bound eEF1A is recycled to the active form by the GEF eEF1B2. Because TIP30 binds eEF1A1 in its middle region where also eEF1B2 binds (Fig 3I), we analyzed whether TIP30 affects the interaction between eEF1A1 and eEF1B2. Increasing concentrations of recombinant TIP30 decreased binding between eEF1A1 and eEF1B2 in a GST-pulldown assay (Fig 4A). Similarly, overexpression of TIP30 in PE-stimulated isolated

---

**Figure 3. TIP30 interacts with eEF1A.**

A  Western blot analysis of GST-pulldown assays with GST-TIP30 fragments of the indicated length (as amino acids from N-terminus) with Myc-tagged binding partners. Red asterisks indicate isolated GST-TIP30 fusion proteins.

B  Western blot analysis of co-immunoprecipitation (IP) from HEK cells transfected with GST-TIP30 and eEF1A1-Myc.

C  Confocal microscopy images of neonatal rat cardiomyocytes (NRCM) stained with antibodies for endogenous TIP30 (green) and endogenous eEF1A1 (red). DAPI: blue (scale bar: 20 μm).

D  Microscopy images of isolated neonatal rat cardiomyocytes and subsequent proximity ligation assay (PLA). Red: endogenous TIP30/eEF1A1 interaction; Blue: DAPI. Control cells were stained for TIP30/Myc-tag interaction (scale bar: 50 μm).

E  Western blot analysis of co-immunoprecipitation (IP) for endogenous TIP30 and eEF1A1 protein in NRCM after stimulation with phenylephrine for 24 h. IgG HC—high chain of IgG molecule.

F  Scheme showing the structure of TIP30-His deletion mutants that were used in GST-pulldown assays in (G). The C-terminal His-tag is highlighted in blue.

G, H  Western blot analysis of GST-pulldown assays with GST or GST-tagged eEF1A1 (GST-eEF1A1) and TIP30-His full-length protein (TIP30-His) and TIP30 deletion mutants ΔN25 (TIP30-ΔN25-His), ΔN52 (TIP30-ΔN52-His), ΔC15 (TIP30-ΔC15-His), and Δ102-107 (TIP30-ΔN102-104-His, G) and Western blot analysis of GST-tagged eEF1A1 and GST-control (H).

I  Scheme depicting eEF1A domains, their different binding patterns for GDP/GTP or eEF1B2 and the His-tagged eEF1A1 mutants. Western blot analysis of GST-pulldown assays with GST-TIP30 and indicated His-tagged eEF1A1 mutants are shown.

J  Quantitative real-time PCR analysis of eEF1A1 and eEF1A2 mRNA abundance in hearts from neonatal mice, adult wild-type mice (adult) and adult wild-type mice 2 weeks after TAC surgery (adult TAC, $N$ = 3–4 mice/group). *$P$ < 0.05, ***$P$ < 0.001. One-way ANOVA with Sidak's multiple comparisons test.

K  Western blot analysis for eEF1A1, eEF1A2, and GAPDH in hearts from wild-type mice at the age of 7 days (7 d) and 8 weeks (8 w) as well as 2 weeks (TAC 2 w) after TAC surgery.

L  Western blot analysis of co-immunoprecipitation (IP) with GST-TIP30 and eEF1A2-His.

Data information: Data are shown as mean ± SEM.
Source data are available online for this figure.

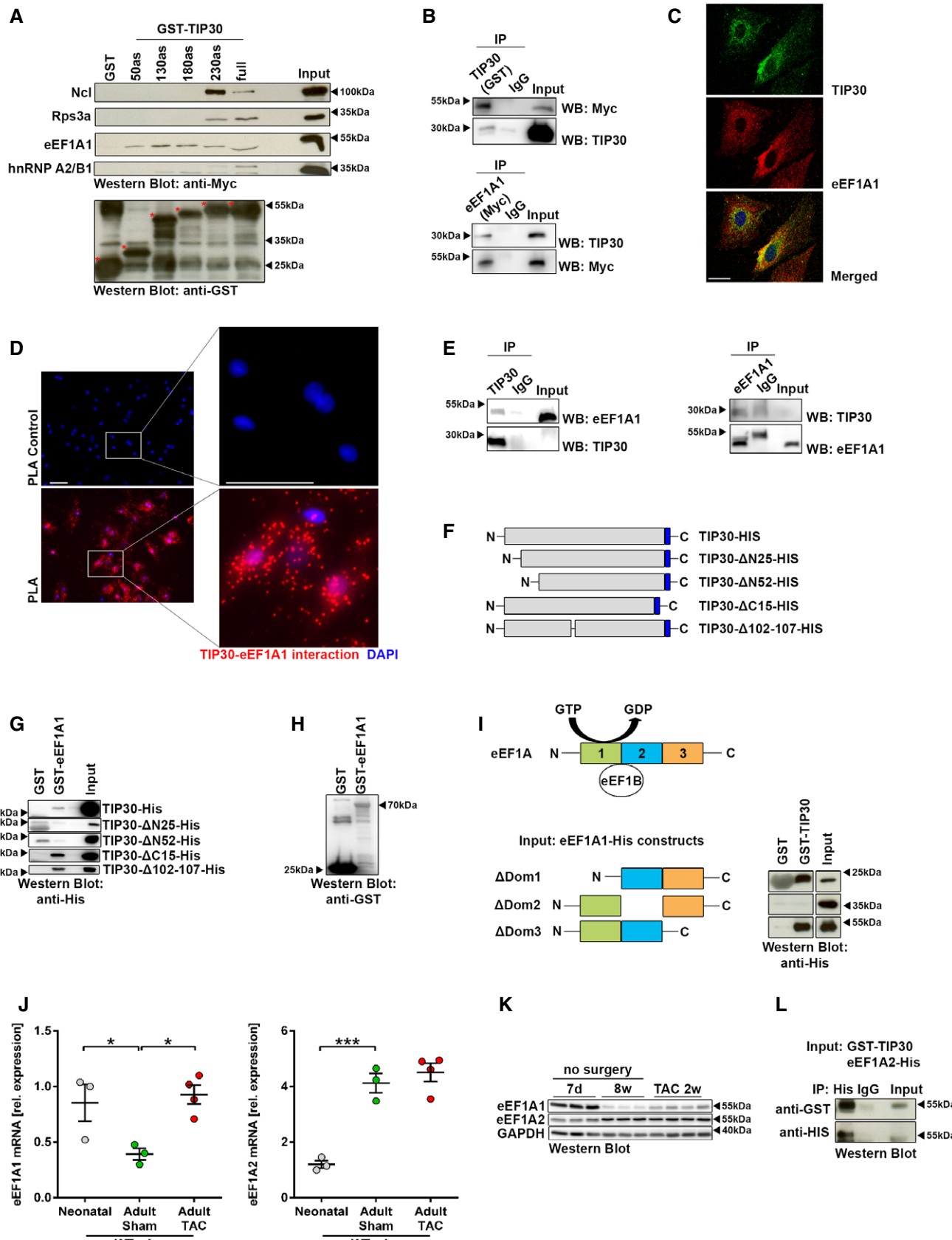

Figure 3.

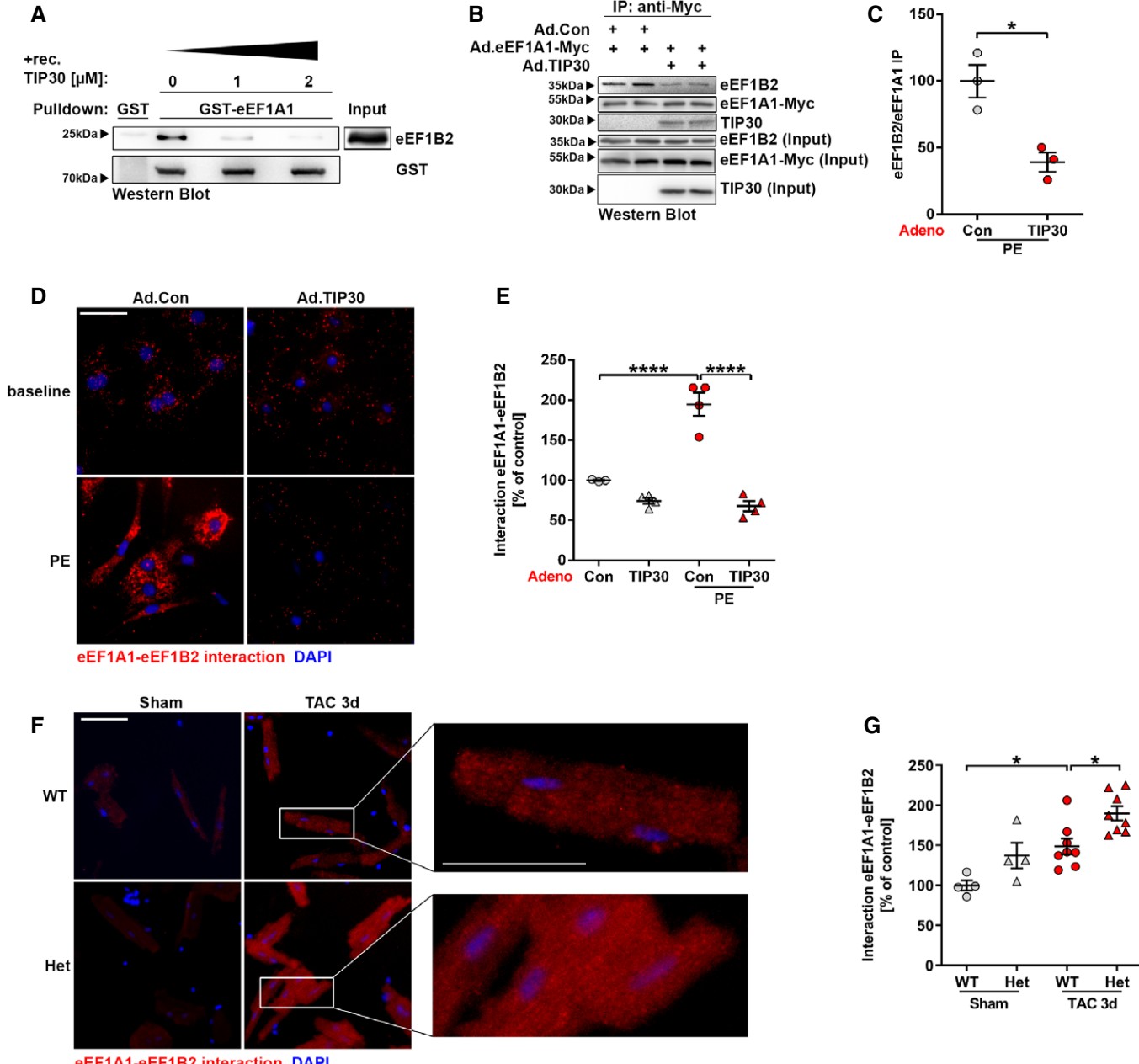

**Figure 4. TIP30 inhibits the interaction of eEF1A1 with its essential co-factor eEF1B2.**

A   Western blot analysis of GST-pulldown assays with GST or GST-eEF1A1 and purified eEF1B2-His. Purified TIP30-His was added in indicated amounts.

B   Western blot analysis of anti-Myc immunoprecipitation (IP) in NRCM co-transduced with Ad.eEF1A1-Myc and either control virus (Ad.con) or Ad.TIP30. Endogenous eEF1B2 was detected. The IP input for eEF1B2, eEF1A1-myc, and TIP30 is shown below.

C   Quantification of eEF1B2 abundance after eEF1A1-myc IP under conditions shown in (B) ($N = 3$ IP's/group). *$P < 0.05$. Two-sided Student's $t$-test.

D   Microscopy images of NRCM after adenoviral transduction with Ad.TIP30 or control virus (Ad.Con) and stimulation with phenylephrine (PE) and subsequent proximity ligation assay (PLA). Red: eEF1A1-eEF1B2 interaction; Blue: DAPI (scale bar: 50 μm).

E   Quantification of eEF1A1-eEF1B2 interaction in conditions described in (D) ($N = 3$–4 samples/group). ****$P < 0.0001$. One-way ANOVA with Sidak's multiple comparisons test.

F   Microscopy images of adult mouse cardiomyocytes isolated from hearts 3 days (d) after TAC or sham surgery and subsequent PLA. Red: eEF1A1-eEF1B2 interaction; Blue: DAPI (scale bar: 100 μm). Inserts represent high magnification of the indicated areas.

G   Quantification of eEF1A1-eEF1B2 interaction in adult mouse cardiomyocytes isolated from hearts 3 days after TAC or sham surgery and subsequent PLA ($N = 4$–6 mice/group). *$P < 0.05$. One-way ANOVA with Sidak's multiple comparisons test.

Data information: Data are shown as mean ± SEM.
Source data are available online for this figure.

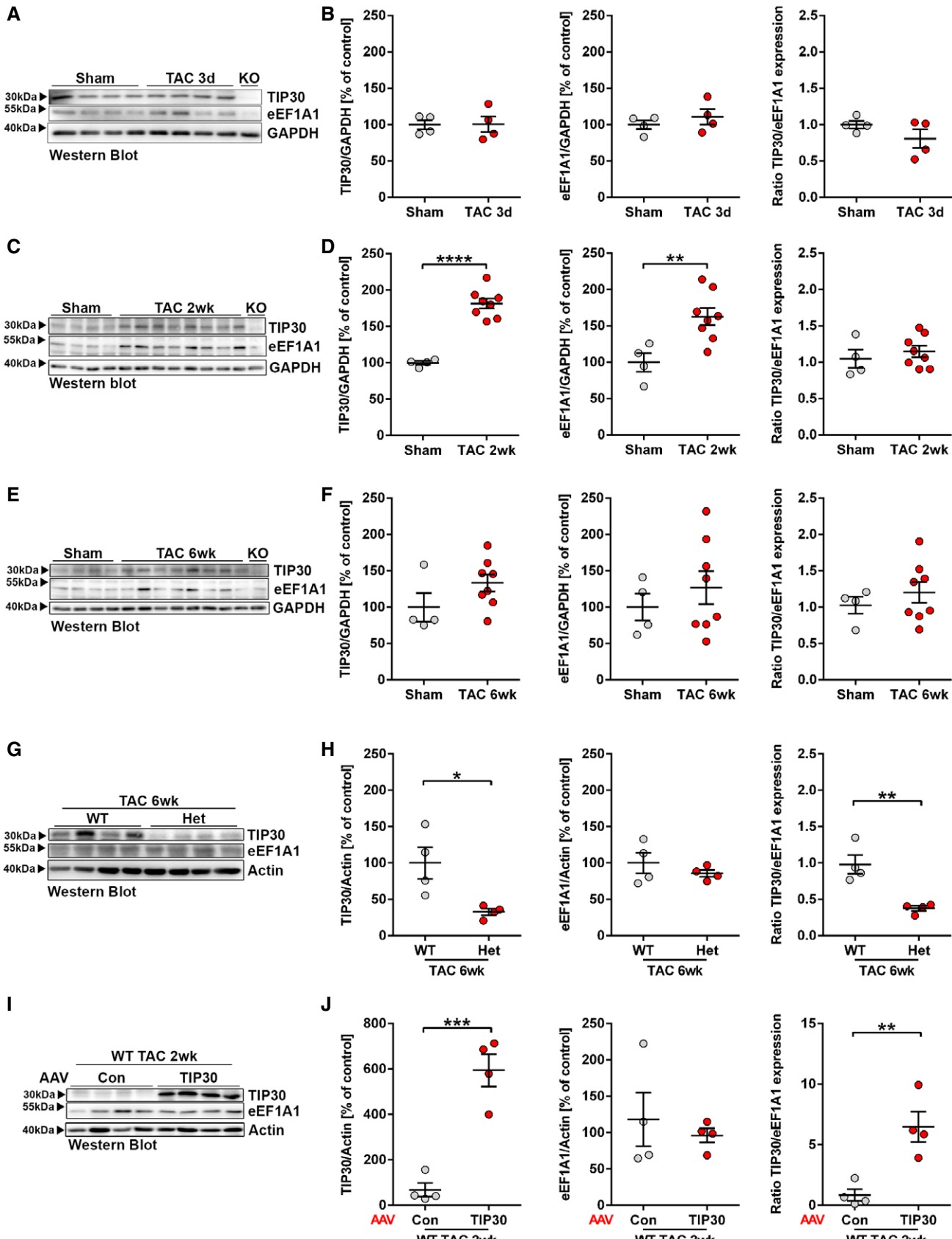

Figure 5.

**Figure 5. A stable TIP30/eEF1A1 ratio is maintained in homeostasis and compensated growth conditions.**

A–F    Western blot analysis for TIP30, eEF1A1, and GAPDH in C57BL/6 WT mice 3 days, $N = 4$ mice/group (A, B), 2 weeks, $N = 4–8$ mice/group (C, D), and 6 weeks, $N = 4–8$ mice/group (E, F) after TAC or sham surgery and their quantification. KO denotes TIP30 homozygous knock-out.

G, H    Western blot for TIP30, eEF1A1, and Actin in TIP30 Het mice 6 after TAC surgery and their quantification ($N = 4$ mice/group).

I, J    Western blot for TIP30, eEF1A1, and Actin 2 weeks after sham or TAC surgery in AAV-Con or AAV-TIP30-treated C57BL/6 WT mice and their quantification ($N = 4$ mice/group).

Data information: Data are shown as mean ± SEM. A ratio of TIP30 and eEF1A1 expression was calculated for each condition. *$P < 0.05$, **$P < 0.01$, ***$P < 0.001$, and ****$P < 0.0001$. Two-sided Student's $t$-test.

Source data are available online for this figure.

cardiomyocytes decreased binding of eEF1B2 to eEF1A1, which instead bound TIP30 under these circumstances (Fig 4B and C). In contrast, TIP30 did not affect tRNA binding by eEF1A1 (Appendix Fig S2). We employed a proximity ligation assay to assess the interaction of endogenous eEF1A1 and eEF1B2 in cardiomyocytes *in situ*. While little interaction was seen in unstimulated cells, pro-hypertrophic stimulation with PE strongly increased binding between both proteins. Strikingly, overexpression of TIP30 completely inhibited increased binding between eEF1A1 and eEF1B2 during PE stimulation (Fig 4D and E). Accordingly, the eEF1A1-eEF1B2 interaction was enhanced by 3 days of TAC treatment in adult cardiomyocytes of WT mice, but was even more exaggerated in cardiomyocytes from Het mice 3 days after TAC (Fig 4F and G). Therefore, the interaction between eEF1A1 and its GEF eEF1B2 increased during hypertrophic stimulation (enabling more translationally active eEF1A1) and this was even more facilitated by reduced TIP30 levels. In turn, elevated TIP30 levels interfered with eEF1A-eEF1B2 binding and thus inhibited recycling of GDP-bound eEF1A to its translationally active GTP-bound form.

## A stable ratio of TIP30/eEF1A1 abundance is maintained in the myocardium during homeostasis and compensated overload, but is reduced in advanced heart failure and hypertrophic cardiomyopathy

Because TIP30 exerts an inhibitory role on eEF1A1 during translational elongation (by interfering with eEF1B2 binding), we analyzed the abundance of TIP30 in relation to eEF1A1 levels. The TIP30/eEF1A1 ratio was maintained at 1 in a rather stable manner after sham surgery as well as 3 days (Fig 5A and B), 2 weeks (Fig 5C and D), and 6 weeks after TAC (Fig 5E and F) in hearts of WT mice. While neither TIP30 nor eEF1A1 were significantly regulated 3 days and 6 weeks after TAC, eEF1A1 levels increased significantly 2 weeks after TAC (versus sham), which was accompanied by a significant increase in TIP30 levels. The cardiac TIP30/eEF1a1 ratio naturally dropped below 1 in TIP30 Het mice after sham and TAC surgery, due to reduced TIP30 levels and virtually unchanged eEF1A1 abundance (Fig 5G and H). When linking the TIP30/eEF1A1 ratios to the degree of cardiac hypertrophy and function during pressure overload (see Fig 1), one could infer that a ratio around 1 might allow the development of moderate hypertrophy with compensated heart function, while a ratio < 1 could enable exaggerated hypertrophy and cardiac dysfunction, presumably because of disinhibition of eEF1A1 due to reduced TIP30 levels. By overexpression of TIP30 via AAV-TIP30, the TIP30/eEF1A1 ratio was increased (Fig 5I and J), which led to reduced cardiac hypertrophy and improved heart function after TAC (Fig 2).

To analyze the TIP30/eEF1A1 ratio in human heart failure, we assessed myocardial mRNA levels of TIP30 and eEF1A1, because TIP30 protein could not be quantified in human samples due to the lack of a specific antibody. The TIP30/eEF1A1 ratio was strongly reduced in human failing hearts (with ischemic or dilated cardiomyopathy) as well as in human hearts from patients with hypertrophic cardiomyopathy (Fig 6A and B, Appendix Table S2). Mice with muscular dystrophy due to the mdx mutation in the dystrophin gene serve as model of human cardiomyopathy. Similar as in human cardiomyopathy, we found a strongly reduced myocardial abundance of TIP30 mRNA and protein and a markedly reduced TIP30/eEF1A1 ratio in the myocardium of 6-month-old mdx mice (Fig 6C, Appendix Fig S3A). To test whether elevation of the TIP30/eEF1A1 ratio could improve cardiomyopathy, we administered AAV-TIP30 to mdx mice. AAV-TIP30-treated mdx mice exerted a significantly reduced left ventricular wall thickness 3 and 9 months after AAV administration versus AAV-control-treated mice (Fig 6D and E). Consistent with reduced wall thickness, the AAV-TIP30-treated mdx mice exerted a significantly lower HW/TL ratio as sign of ameliorated cardiac hypertrophy at the age of 9 months (Fig 6F). Cardiac ejection fraction was unchanged, presumably because beside hypertrophy, cell death and fibrosis also contribute to cardiac dysfunction in mdx mice, but are not addressed by AAV-TIP30 (Yue *et al*, 2003; Bostick *et al*, 2012; Schinkel *et al*, 2012). Still, AAV-TIP30, but not AAV-control-treated mice increased cardiac contractility (Ees, end-systolic elastance, assessed by Millar catheterization) during stimulation with dobutamine (Appendix Fig S3B and C). Although the association of the TIP30/eEF1A1 ratio with the degree of cardiac hypertrophy is in line with our hypothesis, it should be emphasized that a variety of other variables affect the outcome of hypertrophy (e.g., the presence or absence of additional hypertrophic stimuli, the genetic background, and age).

### TIP30 negatively regulates protein synthesis

Because TIP30 interferes with eEF1A1, we hypothesized that it inhibits peptide chain elongation during protein synthesis. When we measured protein synthesis by determining incorporation of the exogenously added aminoacyl-tRNA analogue puromycin into newly synthesized proteins after 3 h of pro-hypertrophic stimulation with PE in isolated cardiomyocytes, we found a markedly increased incorporation of puromycin in cells transduced with Ad.Control, which was strongly reduced in Ad.TIP30-treated cardiomyocytes (Fig 7A and B). This confirmed protein synthesis inhibition by TIP30 overexpression. Polysome profiling from isolated cardiomyocytes showed enhanced polysome formation due to PE stimulation in Ad.Control-infected cells (Fig EV4A and B). Overexpression of

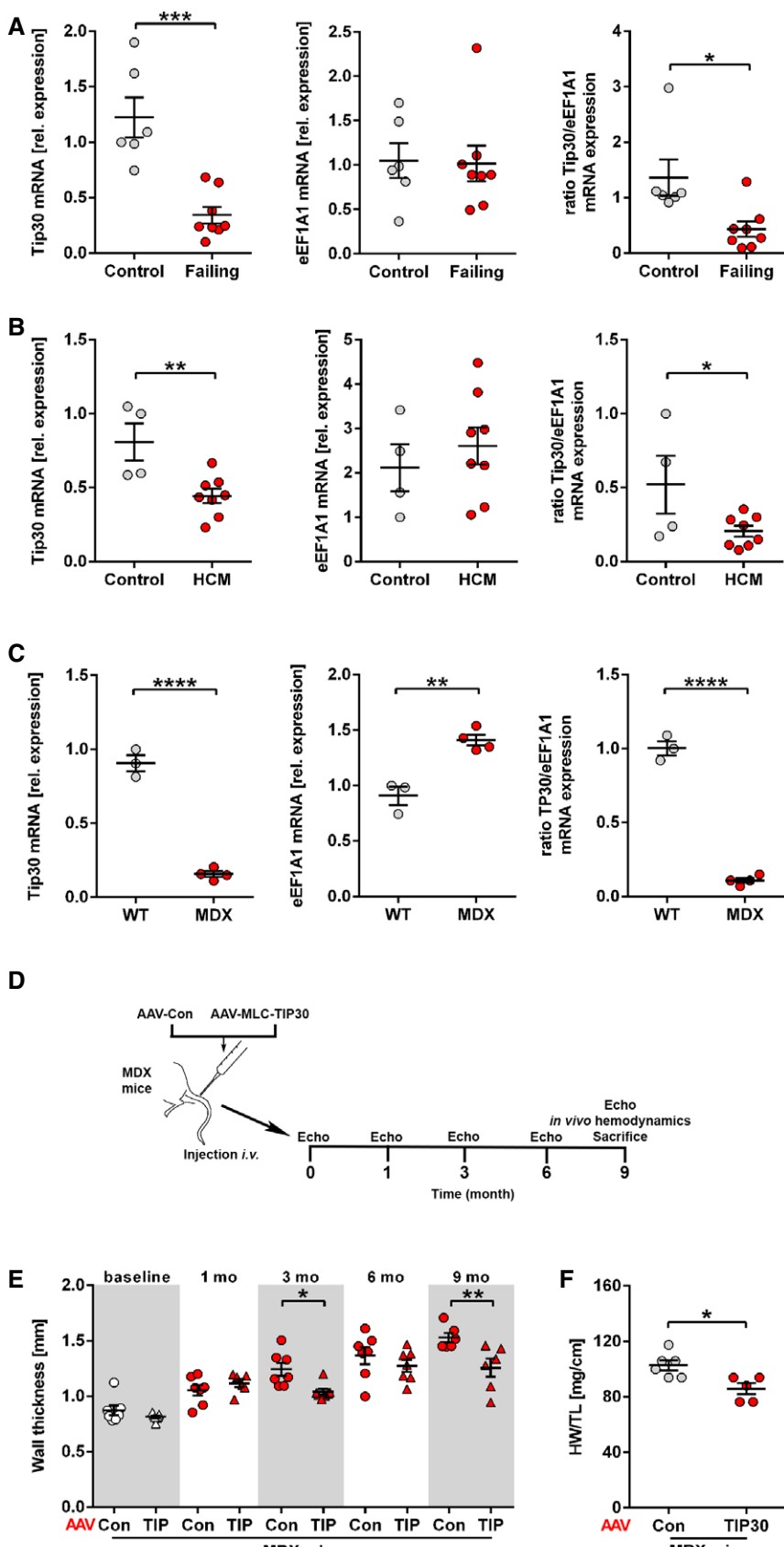

Figure 6.

Figure 6. The TIP30/eEF1A1 ratio decreases in human failing and hypertrophic cardiomyopathy hearts.

A–C    Quantification of Tip30 and eEF1A1 mRNA transcript abundance in human failing hearts, N = 6–8 hearts/group (A), in patients with hypertrophic cardiomyopathy
       (HCM; B, N = 4–8 hearts/group) and 6-month-old mdx mice or WT mice, N = 4 mice/group (C). A ratio of Tip30 and eEF1A1 expression was calculated for each
       condition. *P < 0.05, **P < 0.01, ***P < 0.001, and ****P < 0.0001. Two-sided Student's t-test.
D      Schematic representation of AAV-study design in mdx (MDX) mice.
E      Echocardiographic diastolic left ventricular wall thickness in MDX mice at the age of 2 months at baseline, and 1–9 months (mo) after injection of AAV9-Con or
       AAV9-TIP30 (TIP; N = 6–7 mice/group). *P < 0.05, **P < 0.01. One-way ANOVA with Sidak's multiple comparisons test.
F      Quantification of HW/TL ratio of 9-month-old MDX mice, N = 5–6 mice/group. *P < 0.05. Two-sided Student's t-test.

Data information: Data are shown as mean ± SEM.

TIP30 during PE administration produced a profile characterized by accumulation of 80S ribosomes and a depletion of polysomes. This profile is compatible with inhibition of protein synthesis either at a late step in translation initiation or early in elongation (Schneider-Poetsch *et al*, 2010). To decipher this further, we transfected cardiomyocytes with a luciferase plasmid that reports cap-dependent translation (dependent on functional translational initiation) as renilla luciferase activity and cap-independent translation (i.e., not dependent on functional translation initiation) as firefly luciferase activity (Fig EV4C). In this assay, both renilla and firefly luciferase activities were similarly induced by PE stimulation in control cardiomyocytes and were similarly inhibited by TIP30 overexpression (Fig EV4D and E). This indicated that TIP30 primarily interferes with translational elongation, because inhibition of translational initiation would have reduced selectively only the renilla signal. A 2D-Gel-based comparative proteomic analysis (DIGE) of cardiomyocytes during PE stimulation with and without TIP30 overexpression did not show selective inhibitory effects of TIP30 on the abundance of specific proteins (Appendix Table S3), supporting a role of TIP30 as a general inhibitor of protein synthesis. Furthermore, inhibition of protein synthesis by TIP30 overexpression during TAC induced hypertrophy *in vivo* was confirmed by assessment of cardiac puromycin incorporation (Fig EV4F and G). Next, we assessed protein synthesis in mice with reduced TIP30 levels compared to WT mice. Puromycin labeling indicated enhanced protein synthesis in Het mice, but not KO mice 3 days after TAC (Fig EV5A). Because cardiac overload puts an increased demand on the protein folding capacity of the endoplasmic reticulum (ER), we assessed whether the unfolded protein response (UPR) was initiated due to ER-stress (Arrieta *et al*, 2018). We found that the adaptive UPR response genes Hrd1, Xbp1, and Manf were significantly upregulated in the homozygous TIP30 KO mice after TAC, while only a trend was visible in Het mice after TAC. Rheb

was upregulated in both genotypes after TAC, but more significantly in homozygous KO mice (Fig EV5B).

Hence, increased TIP30 levels inhibited cardiomyocyte protein synthesis, while reduced levels of TIP30 predisposed to increased myocardial protein synthesis during hypertrophic stimulation. Especially in homozygous TIP30 KO mice, pathological overload is associated with activation of the UPR.

**The anti-hypertrophic role of TIP30 depends on eEF1A**

To establish a direct link between eEF1A and cardiomyocyte hypertrophy, we used narciclasine, which acts as inhibitor of eEF1A (Van Goietsenoven *et al*, 2010). Neonatal mouse cardiomyocytes of Het mice exerted more hypertrophy (measured as cell size) compared to WT cells during pro-hypertrophic ET-1 stimulation, but this was completely blunted by narciclasine, suggesting that the enhanced growth of Het cardiomyocytes depended on functional eEF1A (Fig 7C). Even more importantly, narciclasine treatment of WT and Het mice during 2 weeks of pressure overload also blunted increased hypertrophy (measured as HW/TL, wall thickness, and cardiomyocyte area) in Het versus WT mice *in vivo* (Fig 7D–G). At the same time, narciclasine markedly improved cardiac function after TAC in Het mice compared to untreated Het mice (Fig 7H). Myocardial puromycin incorporation was enhanced in the myocardium of Het versus WT mice after TAC (again showing increased myocardial protein synthesis), and this effect was blunted in Het mice receiving narciclasine (Fig 7I).

Finally, we determined whether the effects of TIP30 overexpression depended on eEF1A1: siRNA-mediated eEF1A1 downregulation reduced the anti-hypertrophic effects of overexpressed TIP30 in PE-treated cardiomyocytes (Fig EV5C and D). Similarly, eEF1A inhibition by narciclasine reduced PE-driven cardiomyocyte hypertrophy and strongly ameliorated the growth inhibitory effects of TIP30 expression during PE stimulation (Fig EV5E). Accordingly, while recombinant

Figure 7. TIP30 inhibits protein synthesis in cardiomyocytes.

A      Western blot analysis of isolated neonatal rat cardiomyocytes (NRCM) after adenoviral transduction either with control adenovirus (Con) or Ad.TIP30 (TIP30)
       followed by stimulation with phenylephrine (PE, for 3 h) and puromycin incorporation (for 30 min).
B      Quantification of the Western blot shown in (A) (N = 3 samples/group). *P < 0.05, **P < 0.01. One-way ANOVA with Sidak's multiple comparisons test.
C      Quantification of cell surface area of isolated neonatal mouse cardiomyocyte of *Tip30* Het and WT mice treated with endothelin-1 (ET-1) and narciclasine (Narci)
       or without stimulation as indicated (N = 6 samples/group). *P < 0.05, ***P < 0.001. One-way ANOVA with Sidak's multiple comparisons test.
D      Schematic representation of narciclasine study design.
E–H    Quantification of HW/TL ratio, N = 6–10 mice/group (E), echocardiographic wall thickness, N = 6–10 mice/group (F), cardiomyocyte area, N = 5 mice/group (G), and
       fractional area change (FAC, H, N = 6–10 mice/group) in *Tip30* Het or WT mice 2 weeks after TAC. Animals were treated with narciclasine daily for 14 days after
       TAC as indicated. *P < 0.05, **P < 0.01, ***P < 0.001. One-way ANOVA with Sidak's multiple comparisons test.
I      Western blot analysis of puromycin incorporation in hearts of *Tip30* Het and WT mice 3 days after TAC or sham surgery and daily narciclasine (Narci) injection and their
       quantification (N = 2–4 mice/group). Puromycin was injected 3 h prior to sacrifice. *P < 0.05, **P < 0.01. One-way ANOVA with Sidak's multiple comparisons test.

Data information: Data are shown as mean ± SEM.
Source data are available online for this figure.

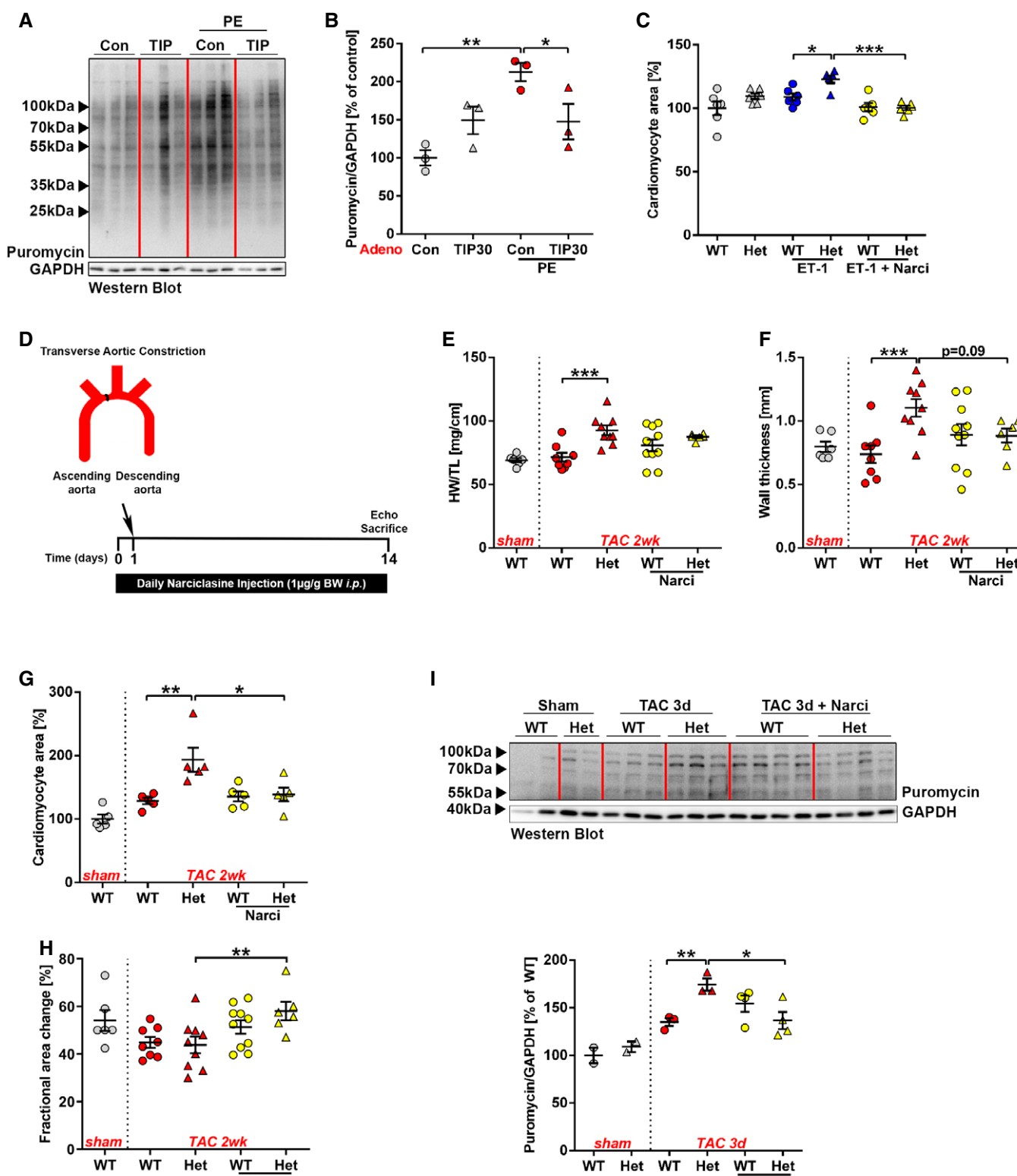

**Figure 7.**

full-length TIP30 blocked translation of a renilla-reporter mRNA in rabbit reticulocyte lysate *in vitro*, a mutant TIP30 (lacking the N-terminal 52 amino acids) devoid of eEF1A1 binding capability, did not significantly inhibit translation in this system (Fig EV5F).

## Discussion

In this study, we found that TIP30 restricts protein synthesis and hypertrophy in cardiomyocytes at the level of translational elongation

by binding to eEF1A1, inhibiting the interaction with its GEF eEF1B2, and thereby reducing the generation of active GTP-bound eEF1A1. While according to our model a balanced abundance of TIP30 and its target eEF1A1 enables compensated cardiac growth during pathological overload, a reduced myocardial TIP30/eEF1A1 ratio like in the late stage of human heart failure promotes exaggerated cardiomyocyte growth and ventricular dysfunction, which could arise from insufficient concomitant myocardial capillary growth. Indeed, heterozygous TIP30 knock-out mice with similarly reduced TIP30 protein levels as we observed in human heart failure, developed enhanced hypertrophy, myocardial capillary rarefaction, and systolic ventricular dysfunction. Overexpression of TIP30 in a cardiomyopathy mouse model with diminished cardiac TIP30 levels, in turn, ameliorated pathological hypertrophy and improved contractility. Besides in heart failure, downregulation of TIP30 was reported previously in aggressive forms of cancer (Ito *et al*, 2003; Zhao *et al*, 2007; Li *et al*, 2009; Tong *et al*, 2009). Not unlike in the heart, reduced TIP30 levels are associated with accelerated growth, cell transformation, and enhanced metastasis formation in human tumors (Ito *et al*, 2003; Zhao *et al*, 2007; Li *et al*, 2009; Tong *et al*, 2009).

We were initially surprised by the fact that heterozygous *Tip30* knock-out mice exerted a stronger cardiac phenotype compared to homozygous KO mice. Quite similarly, however, Het mice also develop more tumors than homozygous KO mice, indicating that TIP30 is haploinsufficient for tumor suppression and for the suppression of heart growth, i.e., that even a reduction of TIP30 levels by about 50% triggers disease (Ito *et al*, 2003). We hypothesize that the complete lack of TIP30 in KO mice induces compensatory mechanisms early during development, which partially ameliorate the consequences of the absence of TIP30. One such mechanism could involve the stronger induction of the UPR as adaptive ER response in homozygous KO mice, which might in part improve heart function during pathological stress by restoring protein homeostasis (Arrieta *et al*, 2018; Blackwood *et al*, 2019; Wang *et al*, 2019). It will be interesting to further decipher the nature of these compensatory mechanisms in future studies, because they might inherit therapeutic potential for hypertrophic heart disease.

Protein synthesis is the main driver of cellular growth, and when cells reach a certain size, mitosis is initiated (Fenton & Gout, 2011). Since adult cardiomyocytes exit cell cycle shortly after birth, hypertrophy is the natural mode of growth in these cells (Heineke & Molkentin, 2006; Hill & Olson, 2008). Indeed, increased protein synthesis has been found to underlie hypertrophic growth during pressure overload in different species (Nagai *et al*, 1988; Imamura *et al*, 1994; Nagatomo *et al*, 1999). Cardiac mechanical overload triggers enhanced mRNA translation within the first days after it has emerged (Ivester *et al*, 1995). Exactly within that time frame of rapid cardiac growth (in our study at day 3 after surgery), reduced TIP30 levels in *Tip30* Het mice led to increased eEF1B2/eEF1A1 association and protein synthesis in response to TAC, indicating that it acts to prevent an overshooting increase in translation and cell growth under these circumstances. TIP30 downregulation or overexpression did not affect cardiomyocyte or cardiac growth without pro-hypertrophic stimulation (e.g., under sham conditions) when short time points were analyzed. We propose that under these conditions, TIP30 only inhibits the low levels of homeostatic protein synthesis that are present in cardiomyocytes not actively growing. Consequently, the effects of TIP30 are not immediately visible, but become important over longer time periods, such as in *Tip30* Het mice, which exert increased heart growth at 7 months of age.

How does TIP30 regulate translation? Unlike, for example, in lung adenocarcinoma cells, where TIP30 directly influences intracellular signaling, this was not apparent in cardiomyocytes (Zhang *et al*, 2011a; Li *et al*, 2013). We rather found that TIP30 associates with the elongation factor eEF1A. Our results indicate that TIP30 binds eEF1A1 at its middle domain, where also aminoacyl-tRNAs and its GEF eEF1B2 interact (Sasikumar *et al*, 2012). While tRNA binding occurred unabated by TIP30, eEF1B2 binding to eEF1A1 was blocked by increased and enhanced by reduced TIP30 concentrations. Since the eEF1B2/eEF1A1 interaction, which was strengthened during cardiomyocyte hypertrophy, is crucial for the propagation of translation (Pittman *et al*, 2009), abrogation of this interaction by TIP30 will stall translational elongation. The regulation of translation at the level of the interaction between eEF1A1 and eEF1B2—like we show here for TIP30—has been previously demonstrated as the result of eEF1B2 phosphorylation by the cell cycle-dependent kinase (CDK)1 leading to reduced interaction with eEF1A1 and downregulation of translation during mitosis (Sivan *et al*, 2011; Sasikumar *et al*, 2012). Reduced activation of translation due to decreased eEF1A1 activity leads to diminished cell proliferation or cell growth in different cell types (Kim *et al*, 2009; Lin *et al*, 2010; Belyi *et al*, 2012). Remarkably, TIP30 appears to act on protein synthesis independent of mTORC1, which was even de-activated in a counter-regulatory manner in TIP30-deficient mice.

We found in this study that TIP30 interacts with both isoforms of eEF1A (eEF1A1 and eEF1A2), which are 92% identical at the amino acid levels and which are thought to fulfill similar functions in the regulation of translational elongation (Abbas *et al*, 2015). We therefore propose that TIP30 acts on both isoforms in a similar manner, although in this study we primarily characterized its effects on eEF1A1. Notably, although eEF1A1 became down- and eEF1A2 upregulated in the myocardium after birth as previously described (Chambers *et al*, 1998), eEF1A1 was strongly re-induced in response to pressure overload. Thus, like in certain forms of cancers, the heart expresses both eEF1A isoforms during overload (Abbas *et al*, 2015). Together, we suggest a new paradigm, whereby cardiomyocyte hypertrophy can be targeted at the level of translational elongation through interference with eEF1A, for example, via overexpression of TIP30 or through substances like narciclasine. Indeed, narciclasine reduced hypertrophy in wild-type rat cardiomyocytes and Het mouse cardiomyocytes *in vitro* as well as in Het mice *in vivo*, although an effect on protein synthesis and heart growth in wild-type mice *in vivo* was not observed here, likely due to insufficient dosing. Especially, since eEF1A is being investigated as novel anti-cancer target (Abbas *et al*, 2015), some of the findings from these studies might be transferable toward the treatment of heart failure in the future.

## Materials and Methods

All mice, reagents, antibodies, plasmids, oligonucleotides, and kits used in this study are summarized in Appendix Table S4.

## Human heart samples

Control tissue was from victims of traffic accidents or from healthy heart organ donors, when the organ was ineligible for transplantation. Samples from failing hearts were derived from patients with ischemic ($n = 2$) or dilated cardiomyopathy ($n = 7$) undergoing cardiac transplantation (Haq *et al*, 2001). Samples from patients with hypertrophic cardiomyopathy (Appendix Table S2) were acquired during myectomy or cardiac transplantation. Their use was permitted by the Massachusetts General Hospital Institutional Review Board (USA), and by the Ethical Committee of the Hannover Medical School, Germany (Az. Z 14.06-A 1871-30724/98 and 2276-2014). Informed consent was obtained from all subjects. The experiments conformed to the principles set out in the WMA Declaration of Helsinki and the Department of Health and Human Services Belmont Report.

## Mouse models

*Tip30* knock-out mice were in a FVBN background and have been described previously (Ito *et al*, 2003). Breeding pairs were a generous gift by H. Xiao (Department of Physiology, Michigan State University, USA). Male *Tip30* WT, Het, and KO mice at 6–8 weeks of age were used for experiments. The mdx mice were previously described (Schinkel *et al*, 2012). AAV9-treated mice used in this study were male C57BL/6N wild-type mice (Charles River Laboratories) as well as mdx mice and *Tip30* Het mice at the age of 6 weeks. The animals had free access to water and a standard diet and were maintained on a 12-h light and dark cycle at a room temperature of $22 \pm 2°C$. For rescue experiments after TAC, narciclasine (#sc-361271, Santa Cruz) was injected into *Tip30* Het mice and WT littermate controls daily for 14 days (1 µg/g BW i.p.). Premature death was a criterion for exclusion from an ongoing experiment. Death rates were not significantly different between experimental TAC groups in this study. All procedures involving the use and care of animals were performed according to the Guide for the Care and Use of Laboratory Animals published by the National Research Council (NIH Publication No. 85-23, revised 1996) and the German animal protection code. Approval was granted by the local state authorities (3.9-42502-04-10/0269 and 33.12-42502-04-15/1871).

## Aortic banding

Transverse aortic constriction (TAC) or sham surgery was performed in 8-week-old mice by subjecting the aorta to a defined 25 gauge constriction as described (Zwadlo *et al*, 2015).

## Transthoracic echocardiography and cardiac catheterization

For echocardiography, mice were anaesthetized with 0.5–1.0% isoflurane and placed on a heating pad to maintain body temperature. Non-invasive, echocardiographic parameters were measured with a linear 30 MHz transducer (Vevo 770, Visualsonics). LV end-diastolic area (LVEDA) and end-systolic area (LVESA) were recorded. Fractional area change was calculated as [(LVEDA − LVESA)/LVEDA] × 100. Intraventricular pressures in mice were assessed in anesthetized (2% isoflurane) and artificially ventilated (MiniVent respirator, Harvard Apparatus) mice using a 1F microtip pressure–volume catheter (PVR 1045, Millar Instruments) coupled with a Powerlab/4SP acquisition system (ADInstruments Ltd), as described (Zwadlo *et al*, 2015). Cardiac parameters were recorded using LabChart (ADInstruments Ltd.) to calculate end-systolic and end-diastolic pressure and heart rate. The hemodynamic measurements in mdx mice were made in closed-chest, spontaneously breathing mice. Mice were anaesthetized by intraperitoneal injection of medetomidine (0.5 µg/g body weight), fentanyl (0.05 µg/g body weight), and midazolam (5 µg/g body weight). A 1.2 Fr catheter (Model FT111B Scisense Inc., London, ON, Canada) was inserted into the left ventricle of the mouse through the carotid artery to simultaneously measure pressure and volumes. Left ventricular volumes were extrapolated from admittance magnitude and admittance phase in real time using the ADVantage PV system (Scisense Inc.). Pressure and volume data were recorded using a Scisense 404—16 Bit Four Channel Recorder with LabScribe2 Software (Scisense Inc.). Transient inferior vena cava compressions were applied to reduce preload and determine end-systolic elastance (Ees). After baseline measurement, intraperitoneal injection of dobutamine with a dose of 20 ng/g body weight was performed and hemodynamic measurements were repeated 5 min after injection.

## Doppler velocity measurements

Following echocardiography, the flow velocity signals of the right carotid artery (RCA) and the left aortic artery (LCA) were measured by placing the Doppler-probe [20 MHz probe of INDUS instruments (version 1.7)] on the right or left side of the cervical midline, respectively (Hartley *et al*, 2011).

## AAV9-mediated overexpression of TIP30

The open reading frame of the *Tip30* gene was cloned into the AAV-vector genome plasmid pds-CMV$_{enh}$-MLC260 or pdsTnT-Cre, respectively. For production of AAV9-TIP30 pseudotyped vectors, these plasmids were used for co-transfection of HEK293T cells together with pDP9rs, a derivate of pDP2rs encoding the AAV9 cap sequence, the AAV2 rep gene, and adenoviral helper sequences. For generation of the control vector AAV9-rLuc (AAV9-control), a vector genome plasmid with Renilla luciferase was used. AAV vectors were produced, purified, and titrated using standard procedures (Werfel *et al*, 2014). AAV9 vectors were administered into the tail vein of 6-week-old male C57BL/6N mice ($5 \times 10^{11}$ vg/ml in PBS) or *Tip30* Het mice ($2 \times 10^{11}$ vg/ml in PBS) 2 weeks prior to TAC and at the age of 2 months in mdx mice.

## Measurement of translation rates *in vivo*

To assess global translation rates in mice, the incorporation of puromycin in actively translated proteins of the heart was measured. Puromycin (Sigma) was injected intraperitoneally (25 mg/kg body weight) into mice 3 days after TAC 3 h prior to sacrifice.

## Primary cardiomyocytes cultures

Neonatal cardiomyocytes were isolated from 1- to 3-day-old Sprague-Dawley rats by Percoll density gradient centrifugation as previously described (Zwadlo *et al*, 2015). Isolated neonatal rat cardiomyocytes (NRCM) were stimulated with either phenylephrine

(20 μM), fetal bovine serum (2%), or endothelin-1 (100 nM) for 48 h to induce hypertrophy. To measure incorporation of puromycin into actively translated proteins 24 h after adenoviral transduction and 3 h of PE stimulation, puromycin (0.1 μg/ml, Sigma) was added into NRCM cell culture for 30 min prior to collecting cell lysates for Western blot analyses. To inhibit eEF1A function, NRCM were incubated with the specific inhibitor narciclasine (100 nm, Santa Cruz) for 24 h as indicated.

To generate Ad.TIP30, rat *Tip30* ORF (NM_001106263.2) was cloned into pShuttleCMV. Virus production was carried out with the AdEasy Adenoviral Vector Systems Kit from Agilent following the protocol. NRCM were transduced the day after isolation. If needed, cells were transfected with siRNA directed against eEF1A1 (SASI_Rn02_00269532, Sigma-Aldrich) or siControl (AM4611, Ambion) using Lipofectamine 2000 (Thermo Fisher Scientific) according to the manufacturer's protocol. Adult cardiac myocytes (ARCMs) were isolated according to AfCS Procedure Protocol ID PP00000125 (http://www.signaling-gateway.org/data/ProtocolLinks.html). Cell size of NRCM and ARCM was measured of at least 100 myocytes per culture dish or mouse using a Zeiss AxioObserver.Z1 Inverted Microscope (Zeiss) and ImageJ (http://rsb.info.nih.gov/ij/). For analysis of protein/DNA ratio, NRCM were pelleted, resuspended in lysis buffer (10 mM Tris, 150 mM NaCl, 4% glycerol, 0.5 mM sodium metabisulfite, 1% Triton X, 0.1% sodium deoxycholate, 0.05% SDS, pH 7.5), and split into aliquots for DNA and protein measurement.

Protein was measured using Micro BCA Protein Assay Kit (Thermo Fisher Scientific) according to the manufacturer's protocol. DNA content was measured using Hoechst 33258 reagent (Sigma-Aldrich) with calf thymus DNA as a standard (Life Technologies). In brief, NRCM suspensions were added to buffer containing 10 mM Tris, 2 M NaCl, 1 mM EDTA, pH 7.4, and 100 ng/ml Hoechst 33258 reagent. Fluorescence was measured using a Modulus Luminometer (Turner BioSystems) at excitation and emission wavelengths of 365 and 460 nm, respectively.

### Measurements of cell contractility and sarcomere length in isolated adult mouse myocytes

Following isolation, ventricular myocytes were placed on 3-cm dishes (#P35G-1.5-10-C, MatTek) coated with laminin (10 mg/cm$^2$) and were washed 3 h later with MEM medium. The isolated myocytes were then transferred to the recording chamber of the IonOptix System. Sarcomere shortening was assessed upon field stimulation (1, 2 and 4 Hz) using a video-based sarcomere length detection system (IonOptix Corporation) at 37°C. The recordings were subsequently analyzed with the Ion Wizard software (IonOptix).

### Quantitative real-time PCR

Total RNA was extracted using TriFast (Peqlab). cDNA was synthesized from 1 μg RNA using Maxima H Minus First Strand cDNA Synthesis Kit (Thermo Fisher Scientific), and quantitative real-time PCR was performed using SYBR Green (Thermo Fisher Scientific) on a MX4000 multiplex QPCR system (Stratagene). Transcript quantities were normalized to GAPDH mRNA with three exceptions where ribosomal protein L7 for normalization of α- and ß-MHC expression (Fig 1H and I) and of Acta1 expression (Fig 2D) was used.

### Immunoblot analysis

For immunoblot (Western blot) analysis, heart samples and isolated cardiac myocytes were lysed in ice-cold lysis buffer (10 mM Tris, 150 mM NaCl, 4% Glycerol, 0.5 mM sodium metabisulfite, 1% Triton X, 0.1% sodium deoxycholate, 0.05% SDS, pH 7.5). Equal amounts of total protein were separated on SDS–polyacrylamide gel electrophoresis under reducing conditions. Densitometry of protein bands was performed using Quantity One software (Bio-Rad).

### Immunostaining

For visualization of specific protein localization in NRCM, cells were stained for anti-TIP30 (#ab177961, Abcam, 1:100) followed by Anti-Rabbit IgG Alexa Fluor® 488 secondary antibody (#4412, NEB, 1:250) and mouse monoclonal anti-eEF1A1 (#sc-21758, Sigma, 1:100) followed by Anti-Mouse IgG Alexa Fluor® 555 secondary antibody (#4409, NEB, 1:250). A goat polyclonal anti-PDGFRα (#AF1062, R&D Systems, 1:100) antibody was used to label cardiac fibroblasts. For visualization of specific protein localization in heart tissue sections after AAV9-TropT-TIP30 transduction, these were stained for anti-TIP30 (#ab71752, Abcam, 1:50) followed by Anti-Rabbit IgG Alexa Fluor® 555 secondary antibody (#4409, Cell Signaling, 1:200) together with FITC-conjugated WGA (#L4895, Sigma-Aldrich). Nuclear staining was performed with VECTASHIELD Mounting Medium (Vector Laboratories) with DAPI. Representative images were acquired using confocal microscopy. Confocal imaging was performed with a TCS SP2 AOBS scan head and an inverted Leica DM IRB microscope equipped with a 63× oil immersion objective. Image analysis was performed using ImageJ.

### Co-immunoprecipitation

After crosslinking with 0.5% PFA/PBS for 5 min and quenching two times with 1.25 M glycine/PBS, cells were lysed in binding buffer (20 mM HEPES pH 7.6, 75 mM NaCl, 0.1% NP-40) supplemented with protease inhibitors (Complete Protease Inhibitor Cocktail, Roche). Lysates were incubated with indicated antibodies or IgG controls overnight at 4°C, incubated with Protein A/G Plus Agarose (Santa Cruz Biotechnologies), washed two times with binding buffer, and resolved using Western blot analysis.

### Proximity ligation assay

Proximity ligation assays were performed in isolated neonatal rat or adult mouse cardiomyocytes using the Duolink In Situ Kit (Olink Bioscience) according to the manufacturer's instructions. Cells were either stained for anti-TIP30 (#ab177961, Abcam, 1:100) and anti-eEF1A1 (#sc-21758, Santa Cruz, 1:100) or anti-eEF1A1 (#ab118703, Abcam, 1:100) and anti-eEF1B2 (#ab77043, Abcam, 1:100). Control cells were stained for anti-TIP30 (#ab177961, Abcam, 1:100) and anti-Myc (#2276, Cell Signaling, 1:100).

### Electron microscopy

Whole hearts of TIP30 WT and Het mice were perfused and fixed in 150 mM HEPES buffer, pH 7.35, containing 1.5% paraformaldehyde and 1.5% glutaraldehyde over night. 2-mm cubes of heart tissue

were then washed in 0.15 M HEPES buffer (2 × 6 min) and 0.1 M cacodylate buffer, pH 7.35 (4 × 6 min), postfixed in 1% osmium tetroxide in cacodylate buffer (2 h), followed by washing steps (4 × 5 min cacodylate buffer, 2 × 5 min water) and 4% aqueous uranyl acetate (over night at 4°C). The heart tissue was then washed in water (2 × 5 min), dehydrated in acetone, and embedded in Epon. 50-nm sections were poststained with 4% uranyl acetate and lead citrate. Electron microscopic examinations were performed by a blinded observer with a FEI Morgagni 268 transmission electron microscope (FEI, Eindhoven, Netherlands) operated at 80 kV using a Veleta CCD camera (Olympus Soft Imaging Solutions).

### Tissue sampling

Excised hearts were arrested in diastole and either embedded in OCT (Tissue-Tek, Sakura) or snap-frozen in liquid nitrogen for further analysis. Hearts in OCT were sectioned at 12 μm thickness and stained with Sirius red according to general protocols to detect collagen fibers. For immunostaining slides were sectioned at 7 μm thickness.

### Luciferase reporter assay

Cell lysates were subjected to a Renilla Luciferase Assay System (Promega) according to the manufacturer's protocol. In brief, 1 day after adenoviral transduction NRCM were transfected with a Renilla luciferase plasmid using GeneTrans II Transfection reagent (MoBiTec). After 48 h, cells were either subjected to a luciferase reporter assay or RNA was isolated for subsequent quantitative real-time PCR. Luciferase activity was detected by using a Modulus Luminometer (Turner BioSystems) and normalized to total protein concentration.

### Cell death assays

To analyze cell death rates, NRCM were transduced with Ad.Control or Ad.TIP30 adenovirus, stimulated with PE for 48 h, washed with 1× PBS, and incubated with 7-AAD (Annexin V: PE Apoptosis Detection Kit I, BD Biosciences) 1:400 in 1× PBS for 15 min. After fixation with 100% ethanol, cells were mounted with VECTASHIELD Mounting Medium (Vector Laboratories) with DAPI. Tissue staining of cleaved caspase-3 (CC3), the activated form of caspase-3, was done with rabbit polyclonal anti-Cleaved Caspase-3 (Asp175, #9661, Cell Signaling, 1:100) using standard procedures.

### *In vitro* translation

*In vitro* translation reactions were performed in Flexi rabbit reticulocyte lysate (Promega) using 50 ng Renilla mRNA as template following the manufacturer's protocol. Purified His-tagged proteins and BSA as control were added in the indicated amounts. The result of translation was analyzed with the Renilla luciferase assay supplied with the reticulocyte lysate.

### Plasmids

The open reading frames of rat *Tip30* (NM_001106263.2), rat *eEF1A1* (BC128723.1), and rat *eEF1B2* (NM_001108799.2) were cloned into the pcDNA3.1/V5-His A vector (Life Technologies) to generate His6-Tag fusion proteins and into the pGEX-4T1 vector (GE Healthcare) to generate GST-Tag fusion proteins. Large deletion mutants of TIP30 as well as eEF1A1 were generated by cloning single fragments of the open reading frames into the indicated vectors. The open reading frames of rat *Ncl* (coding for nucleolin; BC085751.1), rat *Rps3a* (coding for 40S ribosomal protein S3a; BC058483.1), and human *hnRNPA2/B1* (XP_003689535.1) were cloned into pShuttleCMV (Agilent) with a Myc-tag epitope inserted at the C-terminus by PCR.

### Cell culture and transfection of cells

HEK293 and COS-1 cell lines were grown in Dulbecco's modified Eagle medium (DMEM) 4.5 g/l glucose w/o glutamine (Pan-Biotech) supplemented with 10% (*v/v*) of FBS (ATCC) and 1% (*v/v*) of Penicillin/Streptomycin solution (Pan-Biotech) at a temperature of 37°C in the presence of 5% of $CO_2$. Cells were plated in 6-well plates and transfected with Lipofectamine 2000 (Invitrogen) according to the manufacturer's protocol with 1 μg of plasmid DNA. Cells were incubated for 48 h to ensure protein expression.

### Protein purification

His6-tagged proteins were expressed in indicated cell lines and purified with Ni-NTA Spin columns (Qiagen) under native conditions according to manufacturer's protocol. Purified proteins were immediately subjected to Zeba Spin desalting columns (7K MWCO, Thermo Fisher Scientific) to exchange imidazole-containing elution buffer into PBS.

### GST-pulldown assay

GST-tagged proteins were expressed in *Escherichia coli* BL21 (DE3). The purified proteins were bound to glutathione agarose beads (Thermo Fischer Scientific) and incubated for 2 h at 4°C with whole cell lysates or purified proteins as indicated. After washing two times with PBS, bound proteins were eluted by heating the glutathione beads at 95°C for 5 min in SDS loading buffer and detected by Western blot analysis.

### Identification of protein interaction partners after GST-pulldown assay

A GST-pulldown assay was performed with GST or GST-TIP30 from neonatal rat cardiomyocyte lysate. Bound proteins were loaded on a SDS–PAGE under reducing conditions. Protein bands that were apparent only in the GST-TIP30 pulldown lane were excised and identified by mass spectrometry.

### DIGE (difference in-gel electrophoresis) analysis

NRCM were infected with Ad.Control or Ad.TIP30 and were treated with PE (20 μM) on the following day for 24 h. The cells were harvested in DIGE lysis buffer (8 M urea, 30 mM Tris, 4% (*w/v*) CHAPS) with protease inhibitors (Complete Protease Inhibitor Cocktail, Roche). After the determination of protein concentrations, fluorescent dye labeling reactions (GE Healthcare) were conducted and

20 μg of protein was subjected to two-dimensional gel electrophoresis. Three gels per condition were analyzed. Protein spots with more than 1.2-fold significant ($P < 0.05$) difference in abundance between the two conditions were identified by mass spectrometry.

### tRNA binding assay

HEK cells were lysed 48 h after DNA transfection with pcDNA3.1/V5-eEF1A1-His in lysis buffer (50 mM $NaH_2PO_4$, 300 mM NaCl, 10 mM imidazole, 1% Tween). The cell lysate was incubated with Ni-NTA Magnetic Agarose Beads (#36113, Qiagen) according to the manufacturer's protocol. Beads with bound eEF1A1-His were then washed two times with GTP Binding Buffer (25 mM Tris–HCl pH 7.4, 10% glycerol, 75 mM NaCl, 1 mM $Na_3VO_4$, 5 mM NaF, 5 mM β-glycerophosphate, 0.025% Triton X) to remove endogenous bound GDP from eEF1A1. This was followed by incubation with isolated recombinant TIP30-His or BSA as control in indicated concentrations in GTP Binding Buffer. After 30 min, GTP (#A1803, AppliChem) was added to a final concentration of 17 mM and incubated again for 30 min. Finally, 0.5 μl FluoroTect™ Green$_{Lys}$ tRNA (#L5001, Promega) was added. Beads were washed two times with GTP Binding Buffer after 30 min of incubation, and bound eEF1A1-His was eluted with Elution Buffer (50 mM $NaH_2PO_4$, 300 mM NaCl, 500 mM imidazole). Fluorescence of eEF1A1 bound tRNA was measured using a Modulus Luminometer (Turner BioSystems).

### Representative images

Images of histological sections, immunofluorescence pictures, Western blots, and GST-pulldown assays or immunoprecipitations are representative images, and the respective experiments were successfully repeated at least two times.

### Statistics

Statistical analysis was performed using Prism 6 (GraphPad Software). Data are shown as mean ± standard error of the mean (SEM). All experiments were carried out in at least three biological replicates. No statistical method was used to predetermine sample size. Sample size was chosen as a result of previous experience regarding data variability in similar models and experimental set-up. The experiments were not randomized. The investigators were blinded for mouse genotype and treatment during surgeries, echocardiography, cardiac catheterization, organ weight determination, and all histological and immunofluorescence quantifications. The variance was comparable between groups, and normality was assumed. Multiple groups were compared by one-way repeated-measures analysis of variance (ANOVA) followed by Sidak's multiple comparisons test or by unpaired, two-sided Student's $t$-test when comparing two experimental groups. Differences were considered significant when $P < 0.05$. Exact $P$-values and $n$-number for each graph are shown in Appendix Tables S5 and S6.

**Expanded View** for this article is available online.

### Acknowledgements

J.H. was supported by the Deutsche Forschungsgemeinschaft through the Cluster of Excellence Rebirth (EXC 62/1), the Heisenberg Program (HE 3658/6-1; HE3658/6-2), and additional research grants (HE3658/9-1, HE3658/8-1, HE3658/5-1). CT and MG were supported by the Deutsche Forschungsgemeinschaft grant Ga 453/13-1. T.T. was supported by the Cluster of Excellence Rebirth, the EU-funded ERC grant Longheart, and the Foundation Leducq. S.B. was supported by the European commission (FP7-CIG-294278). M.V. was supported by the Deutsche Forschungsgemeinschaft through the Emmy-Noether Program (VO 1659/2-1), additional research grants (VO1659/1-1, VO1659/4-1), and the Baden-Württemberg Stiftung. H.A.K. was supported by the German Center for Cardiovascular Research (DZHK). K.C.W. and J.B. were supported by the Cluster of Excellence Rebirth. O.J.M. was supported by the Deutsche Forschungsgemeinschaft (MU1654/9-1, MU1654/8-1, MU1654/5-1) and Benni&Co (German Duchenne Muscular Dystrophy Foundation). We thank Daniela Bogner for help with animal experiments.

### Author contributions

AGr and JHei initiated and planned the study and all experiments. AGr, OJM, and JHei designed experiments. AGr performed experiments with the help of MS, MK-K, MMM, FAT, US, AGi, JHeg, ER, and CT. SB performed experiments

and analyzed data. TK, AP, and CdR provided human myocardial samples. MV and SD provided important advice, experimental protocols, analyzed data, and critically revised the article. AJ, RB, and OJM provided crucial reagent, performed experiments, and analyzed data. XY and MM performed experiments and analyzed data. AP performed experiments and analyzed data. HX provided the *Tip30* knock-out mice, gave important advice for the project, and revised the article. HAK and JB supported the study and provided infrastructure. MG, TT, HAK, KCW, OJM, and JB critically revised the article. AGr and JHei wrote the paper. JHei supervised the study.

## The paper explained

### Problem

The mammalian adult heart expands in size mainly as a result of cardiac muscle cell growth due to enhanced protein synthesis. During common human diseases such as chronic arterial hypertension, aortic valve stenosis, but also inherited cardiomyopathies, exaggerated heart growth (referred to as hypertrophy) is associated with the development of heart failure and poor prognosis. Cardiac hypertrophy, however, is not sufficiently addressed by current treatment strategies. We hypothesized that endogenous negative regulators of protein synthesis exist in the heart to prevent aggravated hypertrophy and that these regulators could potentially be used for therapeutic purposes in the future.

### Results

We found that the tumor suppressor gene TIP30 counteracts cardiomyocyte growth by binding to the elongation factor 1A (eEF1A). eEF1A is known to promote translation by delivering amino acids to the ribosomes. Our findings suggested that TIP30 through binding to eEF1A reduces protein synthesis by interfering with the interaction between eEF1A and its guanine nucleotide exchange factor eEF1B2, which keeps eEF1A in its active state. Due to this mechanism, even only a reduction of TIP30 by around 50% in heterozygous TIP30 knock-out mice led to increased cardiac protein synthesis, hypertrophy, and cardiac dysfunction in response to pathological pressure overload. TIP30 overexpression, in turn, improved heart function under these circumstances and inhibited myocardial protein synthesis and hypertrophy.

Importantly, TIP30 levels in relation to its target eEF1A are markedly downregulated in failing and cardiomyopathic human hearts, which according to our findings leave eEF1A active and thereby promotes cardiac hypertrophy and failure. Indeed, the phenotype of heterozygous TIP30 knock-out mice during pressure overload could be significantly ameliorated by inhibiting eEF1A with the substance narciclasine. On the other hand, overexpression of TIP30 in cardiomyopathic mice with endogenous TIP30 downregulation potently reduced hypertrophic heart growth over time.

### Impact

We suggest that exaggerated translational elongation is maladaptive during pathological overload. In this regard, reduced TIP30 levels and more active eEF1A in failing hearts might be an interesting new therapeutic target, which could be addressed, for example, by TIP30 overexpression or eEF1A inhibition through substances like narciclasine.

## Conflict of interest

The authors declare that they have no conflict of interest.

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
