## [Review Process File · EMBO Molecular Medicine]

TIP30 counteracts cardiac hypertrophy and failure by inhibiting translational elongation

Andrea Grund, Malgorzata Szaroszyk, Mortimer Korf-Klingebiel, Mona Malek Mohammadi, Felix A. Trogisch, Ulrike Schrameck, Anna Gigina, Christopher Tiedje, Matthias Gaestel, Theresia Kraft, Jan Hegemann, Sandor Batkai, Thomas Thum, Andreas Perrot, Cris dos Remedios, Eva Riechert, Mirko Völkers, Shirin Doroudgar, Andreas Jungmann, Ralf Bauer, Xiaoke Yin, Manuel Mayr, Kai C. Wollert, Andreas Pich, Hua Xiao, Hugo A. Katus, Johann Bauersachs, Oliver J. Müller and Joerg Heineke.

Review timeline:

Submission date:	1 st November 2018
Editorial Decision:	9 th November 2018
Author Appeal Correspondence:	14 th November 2018
Editor Appeal Correspondence:	14 th November 2018
Editorial Decision:	7 th December 2018
Editorial Decision:	27 th June 2019
Revision received:	15 th July 2019
Revision received:	1 st August 2019
Accept:	6 th August 2019

Editor: Celine Carret

Transaction Report:

1st Editorial Decision

9th November 2018

Thank you for submitting your manuscript "TIP30 counteracts cardiac hypertrophy and failure by inhibiting translational elongation" to EMBO Molecular Medicine.

Upon receipt, manuscripts can sometimes be evaluated by the Scientific Editors to deal in a timely fashion with a large number of submissions. In this case, I am afraid that we concluded that your manuscript is not well suited for publication in EMBO Molecular Medicine and have therefore decided not to proceed with peer review.

While potentially of interest to the more immediate community, I am afraid that due to its nature, the article doesn't fit well within EMBO Molecular Medicine as we focus primarily on these studies that provide functional novel insights of clinical and or translational significance in an appropriate model, but also that are conceptually novel and of broad interest. As we do not feel that this is the case here, we therefore cannot offer further consideration to your manuscript.

Author's appeal correspondence

14th November 2018

Thanks a lot for your email and decision, although I must admit that I do not fully understand it. You mentioned that manuscripts "that provide functional novel insights of clinical and or translational significance in an appropriate model, but also that are conceptually novel and of broad interest" would be suitable for Embo Mol Med. Honestly, we feel that our study would fulfill exactly these criteria: We show for the first time an antihypertrophic role for TIP30 in the heart using mouse models (TAC and genetic cardiomyopathy) highly relevant for human heart disease. We also demonstrate the translational relevance of our findings by showing dysregulation of TIP30 in human failing hearts. Perhaps even more importantly, we discover a novel molecular anti-hypertrophic mechanism through interference with translational elongation and we suggest some potential novel therapeutic strategies on this. Therefore we 1) a novel role for TIP30 in the heart 2) provide novel mechanistic insight that could also be relevant for other diseases (such as cancer). TIP30 has not been studied in the heart before and our mechanism of interference between eEF1A and eEF1B is also novel.

We had made very positive experiences with Embo Molecular Medicine previously (see Malek-Mohammadi M et al., 2017, Embo Mol Med); however, recently, especially with your response to EMM-2018-10018, but also with the response to EMM-2018-09956 (although there I agreed with the Editorial Argument that we did not go far enough with regard to a translational perspective on that paper), I now became quite uncertain which papers Embo Mol Med in general might be interested in and whether there still is an interest in cardiovascular biology. This would be important for us to know since we currently have another paper that in principal could be suitable for your Journal, but also with regard to the more distant future, because we are doing mechanistic cardiovascular research with the ultimate aim to identify novel therapeutic approaches.

I hope that you understand my concerns and could help me to figure out better the reasons for your rejection of the TIP30 paper (without even sending it for review) and also to help me understand, which papers Embo Mol Med is looking for,

Editor appeal correspondence

14th November 2018

Dear Prof. Heineken,

Thank you for your letter asking for clarifications. Thanks to your letter I realised that we didn't understand well your paper and after reading it in full I confirm that it is in scope with EMBO Mol Med. Our mistake was to rely on the abstract, which I don't think does a good job at reflecting the content of your paper. I would suggest that you modify it to reflect better the novelty of your findings, but also the use of 2 different models and clinical validation of the findings. I also appreciated that while TIP30 was not associated with any cardiac hypertrophy genetic conditioning as far as I could see, eEF1A was, which further validate the clinical relevance in our view.

I am therefore sending the paper out for review.

I will contact you again upon review completion.

Thank you for the submission of your manuscript to EMBO Molecular Medicine. We have now heard back from the three referees whom we asked to evaluate your manuscript.

You will see that all referees find the data potentially interesting and referee 1 is more supportive than referees 2 and 3, who are more reserved and highlight a few important issues. The main critical point is about causality and referees are not fully convinced that the data demonstrate that tip30 depletion is indeed causing heart hypertrophy. The second critical point is about the underlying mechanism that is not sufficiently developed. They do offer suggestions to improve and strengthen the conclusions and we would like to encourage you to address these comments as indicated.

In particular, during our cross-commenting exercise, it became clear that 1) while the WT TAC was not significantly different from the WT Sham data, ref. saw it as setting a threshold effect, whereby the Het mice with TAC under the same experimental conditions did show a significant increase in HW/TL and other parameters of heart failure. These experimental conditions were therefore judged more than adequate in supporting the author's conclusions that partial loss of TIP30 was pathogenic to the heart. 2) In agreement with reviewer #3, TIP 30 expression needs to be shown for the AAV9 experiment in Figure 1; 3) regarding comment 4 ref#3, should you support the interaction between TIP30 and eEF1A1 with any sort of additional independent assay, the conclusion would be strengthened. 4) please rewrite your abstract keeping in mind to not misrepresent the results of your study.

We would welcome the submission of a revised version within three months for further consideration and would like to encourage you to address all the criticisms raised as suggested to improve conclusiveness and clarity.

REFEREE REPORTS

Referee #1 (Comments on Novelty/Model System for Author):

This is a solid paper and I found the results to be interesting and well done.

Referee #1 (Remarks for Author):

Comments to the Reviewers

In the manuscript by Grund et al., the authors investigate the regulation of protein synthesis and its contribution to the development of cardiac hypertrophy. They characterized the role of the tumor suppressor protein, Tip30, as an inhibitor of mRNA translation by interacting with the eukaryotic translation elongation factor 1A (eEF1A). By modulating Tip30 and eEF1A1 expression in vitro and in vivo, the authors demonstrate that the expression of Tip30 protects against the development of cardiac hypertrophy by inhibiting protein synthesis. Overall, it is a relatively thorough study with supportive data from multiple angles that offers novel insights into the regulation of cardiac hypertrophy by enhanced protein synthesis. While there is a lot of data in the manuscript that supports a unifying hypothesis, several conclusions made by the authors are not completely supported by their data in the present form.

Major comments:

1. Although the in vitro data supports their observations in vivo, it would be helpful to show that there are no baseline differences in cell numbers (e.g. fibroblasts and cardiomyocytes). For example, there appears to be elevated fibrosis in the sham Tip30 het hearts. Similarly, the 7-month-old het mice have enlarged hearts (EV2) - the authors should measure cardiomyocyte size for hypertrophy (and not because there were more cells to begin with).
2. eEF1A1 has also been shown to play critical roles in actin organization and cell proliferation. To what degree did this contribute to the phenotypes observed in the mice? (in other words, does regulating Tip30 expression alter actin organization and proliferation).
3. The authors show that there is an effect in Tip30 het mice but not in Tip30 KO mice and speculate

that this is due to compensatory mechanisms. What happens to eEF1A1 expression in the full KO mice? This is somewhat shown in Figure 5 where it is used as a control sample, where it appears to be significantly ablated. Thus, what happens to protein synthesis in the KO mice?

4. The authors conclude that there was no effect on fibrosis - based on 1 section of PSR staining - this is not adequately quantified.

5. The authors attempt to link the Tip30/eEF1A1 ratio to the degree of hypertrophy - this is rather speculative (especially at the mRNA level) and is not supported by their data - for example the Mdx mouse has the smallest ratio (which supposedly suggests more exaggerated hypertrophy), yet this model has only a weak phenotype at the age studied. Perhaps the authors should temper their speculation here.

6. The Mdx mouse model of cardiomyopathy, at what age was the expression of Tip30 and eEF1A1 assessed? In the methods the authors state that they use 6-week-old Mdx mice. At what age is the HW/TL data measured - at 9 months? Why are their discrepancies in the number of data points (wall thickness vs. HW data)?

7. In Figure 1, a WB to show the overexpression of Tip30 (for M) should be included.

Minor Comments:

1. The authors often use the word 'interestingly' to describe their results. This should be minimized.
2. Why does Narciclasine not reduce protein synthesis in WT mice after TAC?
3. In appendix table S2, m or f should be used to denote sex.
4. In the methods section, the authors state that either mRNA data was normalized to either L7 or Gapdh - as Gapdh co-precipitated with Tip30, it would be important to state where Gapdh was used and that in those cases no changes in the mRNA expression were observed.
5. Why is het capitalized (Het)?

Referee #2 (Comments on Novelty/Model System for Author):

The authors showed convincingly that Tip30 negatively regulates cardiac hypertrophy through its interaction with eEF1A1, which is novel and significant. However, it is unclear to me whether endogenous Tip30 prevents heart failure under stress through suppression of cardiac hypertrophy. Cardiac hypertrophy is not always pathological and Tip30 KO hearts with pressure overload develop heart failure without increases in apoptosis or fibrosis. The level of Tip30/eEF1A1 decreases only at the phase of heart failure when cardiac hypertrophy is already established but cardiac contractility starts to decrease. The authors should further investigate the molecular mechanism of heart failure in the absence of Tip30 since the authors' conclusion is somewhat misleading at the present form. Studying the contraction of isolated cardiomyocytes, protein quality control or myofibrillar structure may allow identifying the true mechanisms. In addition, the quality of the data is insufficient since many important data are missing. In particular, the authors could have shown the level of endogenous Tip30 in the heart and cardiomyocytes at baseline and under hypertrophic stimuli or heart failure.

Referee #2 (Remarks for Author):

The authors demonstrate that Tip30 suppresses cardiac hypertrophy by inhibiting the interaction between eEF1A1 and eEF1B2 and decreasing translational elongation. Tip30 expression was increased in response to hypertrophic stimuli and was decreased in the failing heart. AAV-mediated expression of Tip30 protected the heart from pressure overload in Tip30 het KO mice and the progression of heart failure in MDX mice, while heterozygous knockout of Tip30 exacerbated cardiac dysfunction.

Major comments:

1. Although the authors claim that enhancement of cardiac hypertrophy is the cause of heart failure in Tip30 KO mice, this claim may not be solid. Cardiac hypertrophy is not always pathological and increased wall thickness reduces oxygen consumption and can act protectively in the presence of pressure overload. In fact, apoptosis and fibrosis are not increased. Studying the contraction of isolated cardiomyocytes, protein quality control or myofibrillar structure may allow identifying the true mechanisms. Is the function of individual cardiomyocytes declined? If so, what is the

underlying mechanism?

2. The ratio of Tip30/eEF1A1 did not change during pressure overload (Figure 5), while it was decreased in failing human hearts (Figure 6). These results suggest that the ratio is not critical for the development of TAC-induced pathological hypertrophy, but it is critical only for the development of heart failure. What is the underlying mechanism by which the decreases in Tip30/eEF1A1 promote heart failure?
3. Tip30 homozygous KO mice showed less hypertrophy and better cardiac function compared to heterozygous KO mice after TAC (Figures 1C-F). If Tip30-mediated physical inhibition of the interaction between eEF1A1 and eEF1B2 is critical, I think homozygous Tip30 should be worse than heterozygous KO. Why was homozygous Tip30 KO better than heterozygous KO after TAC?
4. TAC surgery increased dP/dt max and decreased dP/dt min in WT mice (Figures 1G-H). I am wondering whether knocking down Tip30 is really detrimental in the context of pressure overload.

Specific comments:

1. In all TAC experiments, the authors should show the level of LV systolic pressure and the pressure gradient.
2. Appropriate control groups are missing in Figures 1N-O, including WT mice transduced with TropT-TIP30 after Sham and TAC. In these experiment the level of Tip30 in sham operated and TAC with or without rescue should be shown.
3. In EV Figure 1, the authors should show the level of HW/TL after 2 weeks TAC.
4. In Figure 2AB and EF, the authors should show the level of Tip30 with or without hypertrophic stimuli.
5. The quality of immunoprecipitation assay is not high enough in Figures 3B and D. The authors should improve the data.
6. The authors should include the control of full-length TIP30, such as N-terminal truncated TIP30, to exclude the possibility of non-specific inhibition in Figure 4A.
7. Please provide the reason why the ratio of Tip30/eEF1A1 is decreased in hypertrophic cardiomyopathy (HCM). Are the patients end-stage heart failure?

2nd Revision - authors' response

27th June 2019

POINT BY POINT RESPONSES TO EDITOR'S AND REVIEWER'S COMMENTS

Below the reviewer comments appear in bold, followed by our response in normal font.

Editor's comments:

In particular, during our cross-commenting exercise, it became clear that

1) while the WT TAC was not significantly different from the WT Sham data, ref. saw it as setting a threshold effect, whereby the Het mice with TAC under the same experimental conditions did show a significant increase in HW/TL and other parameters of heart failure. These experimental conditions were therefore judged more than adequate in supporting the author's conclusions that partial loss of TIP30 was pathogenic to the heart.

We want to thank the Editor for this supportive statement, with which we fully agree.

2) In agreement with reviewer #3, TIP 30 expression needs to be shown for the AAV9 experiment in Figure 1;

We now included this blot in the new Figure 1P in the revised manuscript. This blot shows only a mild overexpression of TIP30 by AAV9-TropT-TIP30, which was intended, since we wanted no strong overexpression, but rather a replacement of the lost endogenous TIP30 in TIP30 Het mice. Cardiomyocyte specific overexpression is demonstrated in the new EV Figure 1M.

3) regarding comment 4 ref#3, should you support the interaction between TIP30 and eEF1A1 with any sort of additional independent assay, the conclusion would be strengthened.

We verified the interaction of both proteins in cardiomyocytes using a newly established proximity ligation assay employing specific anti-TIP30 and anti-eEF1A1 antibodies. By this means we demonstrate that endogenous TIP30 and eEF1A1 interact in primary cardiomyocytes *in situ*.

4) please rewrite your abstract keeping in mind to not misrepresent the results of your study.

We rewrote our abstract and hope that it is now more suitable.

Reviewers' comments:

Referee #1 (Comments on Novelty/Model System for Author):

This is a solid paper and I found the results to be interesting and well done.

We want to thank this reviewer for her/his positive comments.

Referee #1 (Remarks for Author):

Comments to the Reviewers

In the manuscript by Grund et al., the authors investigate the regulation of protein synthesis and its contribution to the development of cardiac hypertrophy. They characterized the role of the tumor suppressor protein, Tip30, as an inhibitor of mRNA translation by interacting with the eukaryotic translation elongation factor 1A (eEF1A). By modulating Tip30 and eEF1A1 expression *in vitro* and *in vivo*, the authors demonstrate that the expression of Tip30 protects against the development of cardiac hypertrophy by inhibiting protein synthesis. Overall, it is a relatively thorough study with supportive data from multiple angles that offers novel insights into the regulation of cardiac hypertrophy by enhanced protein synthesis. While there is a lot of data in the manuscript that supports a unifying hypothesis, several conclusions made by the authors are not completely supported by their data in the present form.

Major comments:

1. Although the *in vitro* data supports their observations *in vivo*, it would be helpful to show that there are no baseline differences in cell numbers (e.g. fibroblasts and cardiomyocytes). For example, there appears to be elevated fibrosis in the sham Tip30 het hearts. Similarly, the 7-month-old het mice have enlarged hearts (EV2) - the authors should measure cardiomyocyte size for hypertrophy (and not because there were more cells to begin with)

Thank you for these valuable suggestions. We now quantified myocardial fibrosis in the revised manuscript in the different experimental groups (new Figure 1K). Although there was a trend, no significantly increased fibrosis was observed in both groups after TAC, which is in line with our mild TAC approach in this study. Both genotypes were also not different with regard to fibrosis after TAC or sham surgery. We also quantified the number of myocardial fibroblasts after immunofluorescence staining for the fibroblast marker PDGFR α as ratio of fibroblasts/cardiomyocyte. As demonstrated in the new EV Figure 1H-I, there was no significant difference in the number of myocardial fibroblasts between groups. In addition, we analyzed the number of capillary endothelial cells in the heart. As demonstrated in the new Figure 1M-N of the revised manuscript, we found that the capillary/cardiomyocyte ratio as established marker for myocardial capillary density (see Heineke J et al., JCI, 2007; Sano M et al., Nature 2007; Izumiya Y, Hypertension, 2006) strongly dropped in Het mice after TAC. Since capillary rarefaction triggers heart failure (see the above mentioned references), we hypothesize that it contributes to the cardiac dysfunction that we observed in TIP30 Het mice after TAC. Mechanistically, we propose that angiogenesis cannot keep up with the exaggerated growth of the cardiomyocytes in these mice.

As suggested by this reviewer, we also measured the cardiomyocyte cross-sectional area in 7 months old Het and WT mice and found a significantly increased cardiomyocyte cross-sectional area in the Het mice (see new EV Figure 2F-G). This would likely explain the increased heart weight in these mice.

2. eEF1A1 has also been shown to play critical roles in actin organization and cell proliferation. To what degree did this contribute to the phenotypes observed in the mice? (in other words, does regulating Tip30 expression alter actin organization and proliferation).

This is also a very interesting point raised by the reviewer. In this regard, it was suggested by Pittman YR (JBC, 2009; this reference is also included in our manuscript) that "...eEF1B acts as a balancer", because by binding to eEF1A it drives translational elongation and keeps eEF1A away from actin. Therefore, if eEF1A does not bind eEF1B, it is then free to interact with actin and promote actin bundling. Similarly, according to our model, binding of TIP30 to eEF1A replaces binding of eEF1B to eEF1A. Whether the eEF1A-TIP30 complex is still able to bind actin is not known, and we have not investigated this in the current study. Since TIP30 binds to domain II of eEF1A, where also actin binds, the actin binding/bundling function in TIP30 bound eEF1A might also be suppressed. We would like to address this point in future studies. We checked the subcellular structure in Het and WT mice after TAC by electron microscopy (new Expanded View Figure 1K). No major differences were observed, making strong changes in cardiomyocyte actin bundling less likely.

3. The authors show that there is an effect in Tip30 het mice but not in Tip30 KO mice and speculate that this is due to compensatory mechanisms. What happens to eEF1A1 expression in the full KO mice? This is somewhat shown in Figure 5 where it is used as a control sample, where it appears to be significantly ablated. Thus, what happens to protein synthesis in the KO mice?

We cannot fully explain why the homozygous TIP30 KO mice have a milder phenotype than heterozygous KO mice. The expression levels of eEF1A1 do not change significantly in KO mice (please see the Figure under this response). The seemingly reduced eEF1A1 expression in Figure 5 might be the result of an "edge effect" during Western blotting. In the revised version of our manuscript, we provide some evidence that a stronger adaptive Unfolded Protein Response (UPR) to ER stress (likely activated by an initially increased protein synthesis) is induced in KO mice (please see new EV Figure 5B). Since the UPR is adaptive, this could improve cardiac function and ameliorate cardiomyocyte protein synthesis. At the time point, when we measured protein synthesis in mouse hearts 3d after TAC, we could not detect a significantly increased cardiac protein synthesis level in homozygous TIP30 KO mice (please see new EV Figure 5A).

Figure: eEF1A1 protein and GAPDH protein (as loading control) abundance in hearts of the indicated mice.

4. The authors conclude that there was no effect on fibrosis - based on 1 section of PSR staining - this is not adequately quantified.

As stated in response to point 1 by this reviewer, we now quantified fibrosis levels 6 weeks after sham or TAC surgery and found no significant difference between the groups (new Figure 1K).

5. The authors attempt to link the Tip30/eEF1A1 ratio to the degree of hypertrophy - this is rather speculative (especially at the mRNA level) and is not supported by their data - for example the Mdx mouse has the smallest ratio (which supposedly suggests more exaggerated hypertrophy), yet this model has only a weak phenotype at the age studied. Perhaps the authors should temper their speculation here.

We formulated this part of our manuscript more careful. For example we write (page 7 of the revised manuscript, new more careful language is underlined): “When linking the TIP30/eEF1A1 ratios to the degree of cardiac hypertrophy and function during pressure overload (see Fig 1) one could infer that a ratio around 1 might allow the development of moderate hypertrophy with compensated heart function, while a ratio <1 could enable exaggerated hypertrophy and cardiac dysfunction, presumably because of disinhibition of eEF1A1 due to reduced TIP30 levels. By overexpression of TIP30 via AAV-TIP30, the TIP30/eEF1A1 ratio was increased (Fig 5I-J), which led to reduced cardiac hypertrophy and improved heart function after TAC (Fig 2).”

In addition, we added the following sentence at the end of this section (page 8 of the revised manuscript): “Although the association of the TIP30/eEF1A1 ratio with the degree of cardiac hypertrophy is in line with our hypothesis, it should be emphasized that a variety of other variables affect the outcome of hypertrophy (e.g. the presence or absence of additional hypertrophic stimuli, the genetic background, age, etc.).”

6. The Mdx mouse model of cardiomyopathy, at what age was the expression of Tip30 and eEF1A1 assessed? In the methods the authors state that they use 6-week-old Mdx mice. At what age is the HW/TL data measured - at 9 months? Why are their discrepancies in the number of data points (wall thickness vs. HW data)?

The Tip30 and eEF1A1 expression was assessed in 6 months old mdx and corresponding wild-type mice. This information is now added to text in the Results section of the revised manuscript. At 6 weeks of age (as stated in the Methods section), AAV9 was applied to mdx and corresponding wild-type mice for the experiment shown in Figure 6 D-F. The HW/TL ratio was assessed at the age of 9 months, one day after the last echocardiographic examination. This information is now also included in Results section text of the revised manuscript.

For the last echocardiographic measurement, at the age of 9 months 6 mice were analyzed per genotype, with one overlying data point (marked in red). In the night after echocardiography, one mouse in the AAV-TIP30 treatment group died (marked with a *) and the dead body was eliminated by the animal caretaker on the next morning. Therefore, we could not assess the HW/TL ratio in this mouse and the n-number in the AAV-TIP30 group was reduced from n=6 (Figure 6E) to n=5 (Figure 6F).

Wall thickness data at 9 months:

AAV-Con	AAV-TIP30
1,516842	*0,943684
1,447368	1,279474
1,592105	1,088421
1,453158	1,430010
1,707895	1,457018
1,453211	1,337368

HW/TL data at 9 months:

AAV-Con	AAV-TIP30
105,88	94,12
94,12	94,12
100	76,47
105,88	88,24
94,12	76,47
117,65	

7. In Figure 1, a WB to show the overexpression of Tip30 (for M) should be included.

This Western blot is now included in the new Figure 1P of the revised manuscript. This blot shows only a mild overexpression of TIP30 by AAV9-TropT-TIP30, which was intended, since we wanted no strong overexpression, but rather a replacement of the lost endogenous TIP30 in TIP30 Het mice. Cardiomyocyte specific overexpression is demonstrated in the new EV Figure 1M.

Minor Comments:

1. The authors often use the word 'interestingly' to describe their results. This should be minimized.

We have changed this.

2. Why does Narciclasine not reduce protein synthesis in WT mice after TAC?

We assume that the lack of Narciclasine effect on protein synthesis as well as on hypertrophy in wild-type mice was due to insufficient *in vivo* dosing of Narciclasine, which was only administered once per day in mice. In the dose we used, Narciclasine only reduced the over-activation of protein synthesis and heart growth that we observed in Het mice after TAC. We mention this in the revised manuscript in the Discussion section on page 11: *"Indeed, Narciclasine reduced hypertrophy in wild-type rat cardiomyocytes and Het mouse cardiomyocytes in vitro as well as in Het mice in vivo, although an effect on protein synthesis and heart growth in wild-type mice in vivo was not observed here, likely due to insufficient dosing."*

3. In appendix table S2, m or f should be used to denote sex.

We changed this.

4. In the methods section, the authors state that either mRNA data was normalized to either L7 or Gapdh - as Gapdh co-precipitated with Tip30, it would be important to state where Gapdh was used and that in those cases no changes in the mRNA expression were observed.

We added this information to the respective Figure legends.

5. Why is het capitalized (Het)?

In our manuscript, we used the abbreviation/designation WT, for wild-type mice; KO for Tip30 homozygous knock-out mice and Het for Tip30 heterozygous mice. Therefore, all our abbreviations/designations started with a capital letter, and we applied this also to Het.

Referee #2 (Comments on Novelty/Model System for Author):

The authors showed convincingly that Tip30 negatively regulates cardiac hypertrophy through its interaction with eEF1A1, which is novel and significant.

We want to thank this reviewer for her/his positive comments regarding our paper.

However, it is unclear to me whether endogenous Tip30 prevents heart failure under stress through suppression of cardiac hypertrophy. Cardiac hypertrophy is not always pathological and Tip30 KO hearts with pressure overload develop heart failure without increases in apoptosis or fibrosis. The level of Tip30/eEF1A1 decreases only at the phase of heart failure when cardiac hypertrophy is already established but cardiac contractility starts to decrease. The authors should further investigate the molecular mechanism of heart failure in the absence of Tip30 since the authors' conclusion is somewhat misleading at the present form. Studying the contraction of isolated cardiomyocytes, protein quality control or myofibrillar structure may allow identifying the true mechanisms.

This point is well taken, and these are great suggestions by the reviewer. We investigated further why the TIP30 Het mice after TAC develop cardiac dysfunction during the hypertrophic over-growth. We measured single cardiomyocyte contractility using an Ionoptix set up at two weeks after TAC in WT and Het cardiomyocytes, when hearts already display cardiac dysfunction in echocardiography (Expanded View Figure 1A). We measured up to 50 cardiomyocytes in 5 mice per genotype at 3 different stimulation frequencies (1Hz, 2Hz and 4Hz), but could not detect any difference in the sarcomere shortening rate during contraction between both genotypes (please see

the new Expanded View Figure 1G). In addition, we conducted electron microscopy studies in Het and WT mice after TAC, and found no major disturbances or differences in myofibrillar structure between both genotypes (new Expanded View Figure 1K). Therefore, neither a primary cardiomyocyte contractility difference, nor myofibrillar disarray appears to be responsible for cardiac dysfunction in TIP30 Het mice. We also studied the activation of the Unfolded Protein Response (UPR) as response to ER stress, which is viewed as adaptive mechanism during pathological overload (Arrieta et al, *Curr Top Microbiol Immunol*, 2018; Blackwood et al, *Circ Res*, 2019; Wang et al, *Circulation*, 2019). We measured the activation of the UPR by assessing the expression of UPR marker genes (Rheb1, Hrd1, Xbp1, Manf) following the advice of Dr. Shirin Doroudgar, who is an expert in proteostasis research at our University. As shown in the new Expanded View Figure 5B, the UPR response is mainly activated in homozygous TIP30 KO mice, perhaps explaining the milder phenotype in these mice, while this compensatory mechanism is not activated strong enough in Het mice to ameliorate cardiac dysfunction. As potential mechanism to explain the reduced systolic heart function after TAC, we identified capillary rarefaction (like it is also seen in human hearts from patients with end-stage heart failure; Hein S et al., *Circulation*, 2003) in the TIP30 Het mice after TAC: As demonstrated in the new Figure 1M-N of the revised manuscript, we show that the capillary/cardiomyocyte ratio as established marker for myocardial capillary density (see Heineke J et al., *JCI*, 2007; Sano M et al., *Nature* 2007; Izumiya Y, *Hypertension*, 2006) strongly dropped in Het mice after TAC. Since capillary rarefaction leads to cardiac dysfunction (see the above mentioned references) we hypothesize that it contributes to the cardiac dysfunction that we observed in TIP30 Het mice after TAC. Mechanistically, we propose that angiogenesis cannot keep up with the exaggerated growth of the cardiomyocytes in these mice.

In addition, the quality of the data is insufficient since many important data are missing. In particular, the authors could have shown the level of endogenous Tip30 in the heart and cardiomyocytes at baseline and under hypertrophic stimuli or heart failure.

We show the level of endogenous cardiac TIP30 protein at baseline in Figure 1A. Endogenous TIP30 levels at different time points after TAC are displayed in Figure 5 A, C and E. The levels of cardiac Tip30 mRNA levels in human control and failing as well as HCM hearts are presented in Figure 6A and B and in Mdx mice in Figure 6C. We added a Western blot showing the upregulation of endogenous TIP30 protein in isolated rat cardiomyocytes after pro-hypertrophic stimulation with PE, FBS and ET-1 in the new Appendix Figure S1A. Similar to the transient upregulation of cardiac TIP30 after TAC, we interpret this as adaptive cellular response to counteract exaggerated cardiomyocyte protein synthesis and growth.

Referee #2 (Remarks for Author):

The authors demonstrate that Tip30 suppresses cardiac hypertrophy by inhibiting the interaction between eEF1A1 and eEF1B2 and decreasing translational elongation. Tip30 expression was increased in response to hypertrophic stimuli and was decreased in the failing heart. AAV-mediated expression of Tip30 protected the heart from pressure overload in Tip30 het KO mice and the progression of heart failure in MDX mice, while heterozygous knockout of Tip30 exacerbated cardiac dysfunction.

Major comments:

1. Although the authors claim that enhancement of cardiac hypertrophy is the cause of heart failure in Tip30 KO mice, this claim may not be solid. Cardiac hypertrophy is not always pathological and increased wall thickness reduces oxygen consumption and can act protectively in the presence of pressure overload. In fact, apoptosis and fibrosis are not increased. Studying the contraction of isolated cardiomyocytes, protein quality control or myofibrillar structure may allow identifying the true mechanisms. Is the function of individual cardiomyocytes declined? If so, what is the underlying mechanism?

We measured single cardiomyocyte contractility using an Ionoptix set up at two weeks after TAC in WT and Het cardiomyocytes, when hearts already display cardiac dysfunction in echocardiography (Expanded View Figure 1A). We measured up to 50 cardiomyocytes in 5 mice per genotype at 3 different stimulation frequencies (1Hz, 2Hz and 4Hz), but could not detect any difference in the sarcomere shortening rate during contraction between both genotypes (please see the new Expanded

View Figure 1G). In addition, we conducted electron microscopy studies in Het and WT mice after TAC, and found no major disturbances or differences in myofibrillar structure between both genotypes (new Expanded View Figure 1K). Therefore, neither a primary cardiomyocyte contractility defect, nor myofibrillar disarray appears to be responsible for cardiac dysfunction in TIP30 Het mice. We also studied the activation of the Unfolded Protein Response (UPR) as response to ER stress, which is viewed as adaptive mechanism during pathological overload (Arrieta et al, *Curr Top Microbiol Immunol*, 2018; Blackwood et al, *Circ Res*, 2019; Wang et al, *Circulation*, 2019). We measured the activation of the UPR by assessing the expression of UPR marker genes (Rheb1, Hrd1, Xbp1, Manf) following the advice of Dr. Shirin Doroudgar, who is an expert in proteostasis research at our University. As shown in the new Expanded View Figure 5B, the UPR response is mainly activated in homozygous TIP30 KO mice, perhaps explaining the milder phenotype in these mice, while this compensatory mechanism is not activated strong enough in Het mice to ameliorate cardiac dysfunction.

As potential mechanism to explain the reduced systolic heart function after TAC, we identified capillary rarefaction (like it is also seen in human hearts from patients with end-stage heart failure; Hein S et al., *Circulation*, 2003) in the TIP30 Het mice after TAC: As demonstrated in the new Figure 1M-N of the revised manuscript, we show that the capillary/cardiomyocyte ratio as established marker for myocardial capillary density (see Heineke J et al., *JCI*, 2007; Sano M et al., *Nature* 2007; Izumiya Y, *Hypertension*, 2006) strongly dropped in Het mice after TAC. Since capillary rarefaction leads to cardiac dysfunction (see the above mentioned references) we hypothesize that it contributes to the cardiac dysfunction that we observed in TIP30 Het mice after TAC. Mechanistically, we propose that angiogenesis cannot keep up with the exaggerated growth of the cardiomyocytes in these mice.

2. The ratio of Tip30/eEF1A1 did not change during pressure overload (Figure 5), while it was decreased in failing human hearts (Figure 6). These results suggest that the ratio is not critical for the development of TAC-induced pathological hypertrophy, but it is critical only for the development of heart failure. What is the underlying mechanism by which the decreases in Tip30/eEF1A1 promote heart failure?

Indeed, a balanced ration of TIP30 to eEF1A1 (as observed in WT mice after TAC) allows the generation of compensated cardiac hypertrophy, while a reduction of this ratio (during pathological overload) in Het mice facilitates hypertrophy and cardiac dysfunction. As mechanism, we propose that reduced level of TIP30 facilitate the interaction between eEF1B and eEF1A, which is known to drive protein synthesis and cell growth (Pittman YR, *JBC*, 2009). Due to increased cardiomyocyte protein synthesis and overgrowth of these cells in Het mice, vascular growth cannot keep up with the growth of cardiomyocytes resulting in a reduced capillary density, which is also observed in failing human hearts, and which is known to promote cardiac dysfunction.

3. Tip30 homozygous KO mice showed less hypertrophy and better cardiac function compared to heterozygous KO mice after TAC (Figures 1C-F). If Tip30-mediated physical inhibition of the interaction between eEF1A1 and eEF1B2 is critical, I think homozygous Tip30 should be worse than heterozygous KO. Why was homozygous Tip30 KO better than heterozygous KO after TAC?

We were also initially surprised by the fact that homozygous TIP30 KO mice show a milder phenotype than heterozygous mice. This is actually also the case with regard to tumor growth that is observed at older age (in animals > 1 year of age): Heterozygous TIP30 KO mice exert a higher tumor load than homozygous KO mice (). Therefore, TIP30 is haploinsufficient for tumor suppression and for the suppression of heart growth, i.e. even a reduction of TIP30 levels by about 50% triggers disease. Due to the suggestion of this reviewer we also studied the activation of the Unfolded Protein Response (UPR) as response to ER stress, which is viewed as adaptive mechanism during pathological overload and enhanced protein synthesis (Arrieta et al, *Curr Top Microbiol Immunol*, 2018; Blackwood et al, *Circ Res*, 2019; Wang et al, *Circulation*, 2019). We measured the activation of the UPR by assessing the expression of UPR marker genes (Rheb1, Hrd1, Xbp1, Manf) following the advice of Dr. Shirin Doroudgar, who is an expert in proteostasis research at our University. As shown in the new Expanded View Figure 5B, the UPR response is mainly activated in homozygous TIP30 KO mice, perhaps explaining the maintained heart function in these mice, while this compensatory mechanism is not activated strong enough in Het mice to ameliorate cardiac dysfunction.

4. TAC surgery increased dp/dt max and decreased dp/dt min in WT mice (Figures 1G-H). I am wondering whether knocking down Tip30 is really detrimental in the context of pressure overload.

TAC surgery in wild-type mice left the dp/dt max ratio unchanged, which is in agreement with the mild TAC approach we used in this study. In contrast, this ratio significantly drops during TAC in Het mice, as it is characteristic for reduced cardiac contractility.

On the other hand, a more negative dp/dt min ratio indicates that in diastole the pressure drops quicker over time, i.e. that the diastolic function is better. This was observed in wild-type mice after TAC (again in agreement with the mild TAC, in which sometimes even a hyper-contractile state can be observed). However, in the Het mice after TAC, the dp/dt min ratio is significantly less negative, which means that the left ventricle relaxes less quickly in diastole, indicating a reduced diastolic function.

In addition, a reduced cardiac function in TIP30 Het mice after TAC is indicated by a reduced fractional area change as well as by increased pulmonary congestion (increased lung weights) in Het mice in multiple experiments (Figure 1C, D and R; EV Figure 1A) after TAC.

On the other hand, TIP30 overexpression during TAC leads to a sustained increase in left ventricular function 2, 4 and 6 weeks after TAC, indicating that TIP30 is beneficial under these circumstances (Figure 2J).

Specific comments:

1. In all TAC experiments, the authors should show the level of LV systolic pressure and the pressure gradient.

Cardiac catheterization was only performed in a subset of mice 6 weeks after TAC or Sham surgery (i.e. the mice from which the $+dp/dt$ and the $-dp/dt$ values are demonstrated). The systolic intraventricular pressure (Psys) of these mice is now shown in the new EV Figure 1F of the revised manuscript. Psys increases in both TAC groups, although to a lesser extent in Het mice. This is most likely due to the reduced ventricular contractility in Het mice after TAC, because these mice cannot develop so high intraventricular pressure levels.

Unfortunately, we did not measure the transaortic pressure gradient in all mice subjected to TAC, since we did not have access to an appropriate Doppler device when these experiments were conducted in the first place.

In order to rule out a principally different transaortic pressure gradient after TAC, we operated another cohort of WT and Het mice. We estimated the degree of left ventricular pressure overload by measuring the flow velocity signals of the right carotid artery (RCA) versus the LCA by placing the Doppler-probe (20 MHz probe of INDUS instruments (Version 1.7)) on the right or left side of the cervical midline, respectively (Hartley CJ, Am J Physiol, 2011). Based on these measurements, the RCA/LCA flow ratio significantly increased to a similar degree in both WT and Het mice, indicating a similar degree of pressure overload in both groups (shown in the new EV Fig. 1F)

In addition, I would like to mention that all TAC surgeries in this study were conducted by Malgorzata Szaroszyk, who is co-author of this study and the most experienced animal surgeon in our group. She also operated all mice in three other recent studies by our group (Zwadlo C et al., 2015, Circulation; Appari M et al., 2017, Circ Res; Grund et al., 2018, Cardiovascular Research).

2. Appropriate control groups are missing in Figures 1N-O, including WT mice transduced with TropT-TIP30 after Sham and TAC. In these experiment the level of Tip30 in sham operated and TAC with or without rescue should be shown.

We added the requested additional groups of mice. The results are shown in the new Figures 1Q-R of the revised manuscript. We also now demonstrate a Western blot to show expression on TIP30 protein in all experimental groups (new Figure 1P). This blot shows only a mild overexpression of TIP30 by AAV9-TropT-TIP30, which was intended, since we wanted no strong overexpression, but

rather a replacement of the lost endogenous TIP30 in TIP30 Het mice. Cardiomyocyte specific overexpression is demonstrated in the new EV Figure 1M.

3. In EV Figure 1, the authors should show the level of HW/TL after 2 weeks TAC.

Unfortunately, this is not possible, because in this group of mice, only an echocardiography was performed and they were sacrificed 6 weeks after TAC.

4. In Figure 2AB and EF, the authors should show the level of Tip30 with or without hypertrophic stimuli.

We added a Western Blot showing the upregulation of endogenous TIP30 protein in isolated rat cardiomyocytes after pro-hypertrophic stimulation with PE, FBS and ET-1 in the new Appendix Figure S1A. Similar as the transient upregulation of cardiac TIP30 after TAC, we view this as adaptive cellular response to counteract exaggerated cardiomyocyte protein synthesis and growth.

5. The quality of immunoprecipitation assay is not high enough in Figures 3B and D. The authors should improve the data.

We repeated all relevant immunoprecipitations using a cross-linking protocol. The resulting new blots are shown in the new Figures 3B and E. In addition, we established a new proximity ligation assay to demonstrate the interaction of endogenous TIP30 and eEF1A1 *in situ* (new Figure 3D).

5. The authors should include the control of full-length TIP30, such as N-terminal truncated TIP30, to exclude the possibility of non-specific inhibition in Figure 4A.

This is also a very good suggestion by this reviewer. We aimed to repeat the assay with deltaN-TIP30, but unexpectedly could not produce the His-tagged version of the protein, which was previously easy to produce. After several failed trials, we realized only very close to the re-submission deadline of our manuscript that the respective plasmid DNA was degraded. We therefore have to produce new plasmid, before we can start another round of transfection and protein production, before the GST-pull down experiment can be performed. We will keep this work going to eventually have it ready for a second revision.

On the other hand, we feel that by providing several different experimental set-ups to show that TIP30 interferes with the interaction of eEF1B and eEF1A1 (GST-pull down, Figure 4A; co-IP from cardiomyocytes, Figure 4B-C; proximity ligation assay *in situ* with TIP30 gain and loss of function studies, Figures 4D-G) a non-specific inhibition is unlikely. In addition, we show that expression of the deltaN-TIP30 mutant in rabbit reticulocyte lysate, in contrast to WT Tip30, cannot inhibit translation of Renilla mRNA in this system (EV Figure 5F).

7. Please provide the reason why the ratio of Tip30/eEF1A1 is decreased in hypertrophic cardiomyopathy (HCM). Are the patients end-stage heart failure?

Our study was not designed to study the mechanism of cardiac TIP30 downregulation in end-stage heart failure in humans, in cardiomyopathic Mdx mice or in human HCM patients. The HCM patients were not in end-stage heart failure, as most of them (the HOCM patients) were operated to remove the obstruction from their left ventricular outflow tract. Why TIP30 is downregulated in these different cardiac conditions will be the subject of future studies in our lab.

Referee #3 (Comments on Novelty/Model System for Author):

The authors failed to provide objective data to substantiate their claims that (i) depletion of TIP30 aggravates the detrimental effects of pathologic cardiac hypertrophy

We show in the manuscript in multiple experiments (Figure 1B, 1Q) that heterozygous TIP30 knock-out mice (as well as homozygous TIP30 knock out mice) exert an increased HW/TL ratio, i.e. a bigger heart. We also show that Het mice have bigger cardiomyocytes after TAC (Figure 1L). In addition we show by multiple approaches (echocardiography, cardiac catheterization, increased pulmonary congestion leading to heavier lungs) that Het mice after TAC have a reduced cardiac function (please see Figure 1C, D, F, G, Q, R and EV Figure 1A).

and that (ii) TIP30 interacts and regulates eEF1A1, which, in my opinion, invalidate their conclusions.

We repeated all relevant immunoprecipitations using a cross-linking protocol. The resulting new blots are shown in the new Figures 3B and E. In addition, we established a new proximity ligation assay to demonstrate the interaction of endogenous TIP30 and eEF1A1 *in situ* (new Figure 3D).

Referee #3 (Remarks for Author):

In this manuscript, the authors report data suggesting that myocardial depletion of the tumor suppressor TIP30 worsens, while systemic administration of AAV9 vector encoding a recombinant TIP30 directed by troponin T promoter partially reversed the detrimental effects of load-induced cardiac hypertrophy. Pulldown assays performed with GST-TIP30 and total lysates of NRCM, coupled to MS, show that TIP30 associates with several proteins identified in the translational apparatus. Further experiments focused on the association of TIP30 with eEF1A1 indicated that TIP30 might negatively interfere in the binding of the GEF cofactor eEF1B2 to eEF1A1, inhibiting the protein synthesis invoked by pro-hypertrophic stimuli in NRCM. Besides, the authors show a reduced TIP30/eEF1A1 ratio in the MDX murine model and samples of diseased human hearts, supporting the notion that a reduction of TIP30 may play a role in the pathogenesis of heart failure.

In general, the manuscript is very interesting and has potentially important novelties. However, some conclusions require further supportive evidence and additional controls.

We appreciate the positive comments by this reviewer.

Major points

1) The authors draw a conclusion about the detrimental cardiac effects of TIP30 depletion based on comparisons of the impact of pressure-overload induced by TAC in the structure and function of the left ventricle of wild-type, heterozygous and homozygous TIP30 KO mice. Data shown in Figure 1 (C, E, G and H) indicate a greater deleterious impact of TAC in the heterozygous TIP30 KO mice. However, no statistical differences were seen regarding HW/TL, left ventricle area shortening/tibia length and Δ dp/dt between WT-TAC and sham-operated mice, implying that WT-TAC mice failed to develop (on average) hypertrophic growth in response to pressure overload. These data invalidate the conclusion that depletion of TIP30 aggravates the pathologic hypertrophy in the heterozygous or homozygous TIP30 KO mouse models.

As pointed out by the Editor, we set “a threshold effect, whereby the Het mice with TAC under the same experimental conditions did show a significant increase in HW/TL and other parameters of heart failure. These experimental conditions were therefore judged more than adequate in supporting the author's conclusions that partial loss of TIP30 was pathogenic to the heart.”

In addition, Figure 1Q in the revised manuscript shows a significant increase in the HW/TL ratio between WT mice after TAC versus sham surgery. Het mice also show a significant increase in HW/TL in this setting versus WT mice after TAC.

The successful TAC surgery in our mice was also verified by a significant increase in the systolic ventricular pressure after TAC (new EV Figure 1F), as well as by Doppler measurements to infer a significant and similar pressure overload in TIP30 Het and WT mice after TAC (new EV Figure 1E).

2) Left ventricular sections shown in Figure 1K are not consistent with data of Figure 1C (see comments above). Besides, one cannot conclude about myocardial fibrosis by Figure 1K, as stated on Page 04, lines 26-27.

Although not quite significant, we saw on average bigger hearts in WT mice after TAC, which is demonstrated in Figure 1J of the revised manuscript. We now quantified

myocardial fibrosis in the revised manuscript in the different experimental groups (new Figure 1K). Although there was a trend, no significantly increased fibrosis was observed in both groups after TAC, which is in line with our mild TAC approach in this study. Both genotypes were also not different with regard to fibrosis after TAC. We also quantified the number of myocardial fibroblasts after immunofluorescence staining for the fibroblast marker PDGFR α as ratio of fibroblasts/cardiomyocyte. As demonstrated in the new EV Figure 1H-I, there was no significant difference in the number of myocardial fibroblasts between groups.

3) Data shown in Figure 1N and 1O suggest that cardiac-specific expression of a recombinant TIP30 may rescue the detrimental effects related to its depletion in heterozygous TIP30-KO mice. However, essential for any conclusion of the data is the demonstration that the recombinant TIP30 protein expression is indeed increased in cardiomyocytes following the systemic administration of the AAV9-Tip30 vector. The immunofluorescence imaging data shown in EV Fig 1F are not sufficiently strong evidence in favor of an efficient expression of the recombinant TIP30 in the myocardial cells. Also, as an appropriate control, the authors need to show that an irrelevant protein, equally expressed in cardiomyocytes, would not rescue the detrimental effects of TIP30 depletion in heterozygous mice.

We isolated adult cardiomyocytes from AAV-control as well as AAV-TropT-TIP30 treated mice and analyzed TIP30 overexpression by Western blotting: Indeed, TIP30 was mildly overexpressed in these cells after AAV-TropT-TIP30 treatment (see new EV Figure 1P). This was intended, since we wanted no strong overexpression, but rather a replacement of the lost endogenous TIP30 in TIP30 Het mice. Cardiomyocyte specific overexpression is demonstrated in the new EV Figure 1M.

AAV9-rLuc (AAV9-control) was used as AAV9-control vector over-expressing Renilla luciferase as irrelevant protein that we administered to Het and WT mice before sham or TAC surgery. We also added additional control groups to this experiment (please see the new Figure 1Q-R).

4) The authors performed pull-down assays with recombinant GST-TIP30 and extracts of neonatal cardiomyocytes, coupled to mass spectrometry, for identification of potential partners of TIP30. They report interactions with proteins linked to translational machinery. Some of the protein-protein interactions were verified (Figure 3A and 3B) in cells overexpressing Myc- and GST-tagged recombinant proteins using pull-down and immunoprecipitation assays. However, these biochemical and immunological techniques are unsuitable to confirm the physical interaction between proteins. A pull-down experiment with cell lysates either with endogenous or overexpressed recombinant proteins might contain false positives. The vast majority of the interacting proteins reported in this manuscript (Appendix Table S1) are common high abundance contaminants in pull-down coupled to MS experiments, including eEF1A1. The authors need to provide further in vitro (e.g. fluorescence titration experiments or size exclusion chromatography combined with 'on-line' multi-angle laser light-scattering), and in vivo (e.g. FRET) to confirm the physical interaction between TIP30 and eEF1A1.

These are great suggestion by the reviewer. Unfortunately, in the shortness of time given for the review, we could not find a collaborator performing the suggested interesting new techniques.

To strengthen our data on the interaction of TIP30 and eEF1A1, we repeated all relevant immunoprecipitations using a cross-linking protocol. The resulting new blots are shown in the new Figures 3B and E.

Importantly, we established a new proximity ligation assay to demonstrate the interaction of endogenous TIP30 and eEF1A1 in situ in primary cardiomyocytes (new Figure 3D).

Thank you for the submission of your revised manuscript to EMBO Molecular Medicine. We have now received the enclosed reports from the referees that were asked to re-assess it. As you will see the reviewers are now supportive and I am pleased to inform you that we will be able to accept your manuscript pending editorial final amendments.

REFeree REPORTS

Referee #1 (Comments on Novelty/Model System for Author):

The revised manuscript is now acceptable. I believe the study to be well done and important

Referee #1 (Remarks for Author):

Revisions were adequate and have addressed my concerns

Referee #2 (Comments on Novelty/Model System for Author):

The authors have addressed my concerns.

Referee #2 (Remarks for Author):

The authors have addressed my concerns. The paper is ready for publication.

3rd Revision - authors' response

1st August 2019

Reviewers' comments:

Referee #1 (Comments on Novelty/Model System for Author):

The revised manuscript is now acceptable. I believe the study to be well done and important

Referee #1 (Remarks for Author):

Revisions were adequate and have addressed my concerns

Thank you, we appreciate these statements.

Referee #2 (Comments on Novelty/Model System for Author):

The authors have addressed my concerns.

Referee #2 (Remarks for Author):

The authors have addressed my concerns. The paper is ready for publication.

Thank you, we appreciate these statements.

Corresponding Author Name: Joerg Heineke

Manuscript Number: EMM-2018-10018-V3